# Waste Rubber Recycling: A Review on the Evolution and Properties of Thermoplastic Elastomers

**DOI:** 10.3390/ma13030782

**Published:** 2020-02-08

**Authors:** Ali Fazli, Denis Rodrigue

**Affiliations:** Department of Department of Chemical Engineering, Université Laval, Quebec, QC G1V 0A6, Canada; ali.fazli.1@ulaval.ca

**Keywords:** rubber, recycling, waste polymers, thermoplastic elastomer, compatibilisation

## Abstract

Currently, plastics and rubbers are broadly being used to produce a wide range of products for several applications like automotive, building and construction, material handling, packaging, toys, etc. However, their waste (materials after their end of life) do not degrade and remain for a long period of time in the environment. The increase of polymeric waste materials’ generation (plastics and rubbers) in the world led to the need to develop suitable methods to reuse these waste materials and decrease their negative effects by simple disposal into the environment. Combustion and landfilling as traditional methods of polymer waste elimination have several disadvantages such as the formation of dust, fumes, and toxic gases in the air, as well as pollution of underground water resources. From the point of energy consumption and environmental issues, polymer recycling is the most efficient way to manage these waste materials. In the case of rubber recycling, the waste rubber can go through size reduction, and the resulting powders can be melt blended with thermoplastic resins to produce thermoplastic elastomer (TPE) compounds. TPE are multi-functional polymeric materials combining the processability of thermoplastics and the elasticity of rubbers. However, these materials show poor mechanical performance as a result of the incompatibility and immiscibility of most polymer blends. Therefore, the main problem associated with TPE production from recycled materials via melt blending is the low affinity and interaction between the thermoplastic matrix and the crosslinked rubber. This leads to phase separation and weak adhesion between both phases. In this review, the latest developments related to recycled rubbers in TPE are presented, as well as the different compatibilisation methods used to improve the adhesion between waste rubbers and thermoplastic resins. Finally, a conclusion on the current situation is provided with openings for future works.

## 1. Introduction

Rubber, as an elastomeric material, has the ability of reversible deformation (between 100 up to 1000%), which is significantly influenced by its chemical structure and molecular weight (MW). Ideally, rubber chains should return to their original shape after removing the applied force (stress). The macromolecular chains of rubber are long and oriented without large substituents, which makes them capable of moving and rotating around chemical bonds at low temperatures because of their low glass transition temperature (Tg). Increasing irregularities in the polymer chains or the presence of large substituents (styrene-butadiene rubbers (SBR)) leads to higher rubber Tg.

The production of high-quality rubber at a large scale with a low cost substantially increased with the development of efficient vulcanisation processes. Vulcanisation is defined as the irreversible crosslinking reaction via curing agents (sulfur or peroxide materials) to form a three-dimensional (3D) network between the rubber macromolecules. Several parameters must be controlled in the rubber vulcanisation process such as curing time, temperature, and fillers having a direct effect on the chemical, mechanical, and physical properties of crosslinked rubbers. Incorporation of vulcanizing agents into an unsaturated rubber improves the rubber strength due to the crosslinked structure created. Therefore, vulcanized rubber as an elastic, insoluble, and infusible thermoset material cannot be directly reprocessed. This is an important limitation for material recycling, especially after the end of life of a part. Depending on the final application, different rubbers are mixed with different components and additives. For instance, stabilisers, anti-oxidants, and anti-ozonants are being used in rubber formulation to make tires extremely resistant to severe outdoor conditions (chemical reagents, high temperatures, radiations, and shear stress) during their lifetime [1,2,3].

Tires as the main application of rubber industries are complex materials containing several components suitable to operate in a wide range of environment. Rubber is the main component used for tire manufacturing, which can be classified into natural rubber (NR), SBR, nitrile-butadiene rubber (NBR), and ethylene–propylene–diene-monomer rubber (EPDM). However, the presence of reinforcing fillers, antioxidants, antiozonants, and curating agents in tire formulation makes them resistant to biodegradation, photochemical decomposition, and high temperatures [2,4]. Therefore, waste tires’ management is an important issue with respect to the global growth of tire industries. This paper reviews the progress of waste tire recycling focused on melt blending of ground tire rubber (GTR) with thermoplastic matrix. Furthermore, this review presents developments in surface modification and devulcanisation of GTR and compatibilisation of thermoplastic elastomer (TPE) blends to improve the interfacial adhesion of GTR and thermoplastic matrix.

### 1.1. Microstructural Composition

#### 1.1.1. Elastomers

NR is extensively used in rubber production as an elastomer component. NR with high MW and long chain branches has the ability to crystallize quickly under stretching, leading to high tensile strength and tear growth resistance. Usually, NR is mixed with other synthetic rubbers such as butadiene rubber (BR), hydrogenated nitrile-butadiene rubber (HNBR), SBR, NBR, and EPDM to further improve its properties (tensile strength and tear growth resistance) in tire manufacturing [5].

#### 1.1.2. Fillers

Different fillers such as carbon black (CB), precipitated silica, and clay have been used in rubber formulation to improve the rubber strength. This is done via the formation of a flexible filler network and strong polymer-filler interactions [6]. Stiffening fillers (CB and silica) improve rubber stiffness, tensile and tear strength, hardness, and rupture modulus as a result of increased chain entanglements and shear strength between the polymer chains. Montmorillonite, synthetic mica, and saponite are clay-based fillers used in rubber production due to better mechanical properties’ improvement compared to CB [7]. For example, Okada [8] reported the positive effect of 10 vol.% of organoclay in NBR to achieve similar tensile strengths as rubber formulations with 40 vol.% CB. However, the rubber microstructure might be affected by the size, shape, and molecular structure of the fillers [9].

#### 1.1.3. Other Additives

Several materials have been used to increase the durability and accelerate the crosslinking reaction of rubber compounds. For instance, zinc oxide has been used as an activator during vulcanisation. Mild extract solvate (MES), naphthenic oil, treated distillate aromatic extract (TDAE), and paraffinic oils are being used to improve the rubber’s processability [9]. Nevertheless, the type and level of filler addition strongly depend on the rubber matrix being used.

### 1.2. Rubbers Types

Rubbers can be categorized into different groups: saturated/unsaturated, natural/synthetic, etc. However, according to the application and properties required, there are general rubbers and special rubbers. General rubbers are relatively low-cost materials produced and consumed in large volume, while special rubbers have special properties such as thermal stability/fire resistance, aging resistance, chemical resistance, and swelling resistance in non-polar oils, as well as their elastic properties. Some of the most used rubber materials in industries are described to get a better understanding of their properties and applications.

#### 1.2.1. Natural Rubber 

NR is a biopolymer based on cis-1,4-polyisoprene with a vegetable origin obtained from *Hevea brasiliensis* (Figure 1). NR is an unsaturated rubber with long, regular, flexible, and linear macromolecules, as well as high elastic properties (Tg ~ −70 °C). Unvulcanised NR can be reversibly elongated under high deformation up to 800–1000% due to its highly resilient characteristics. Although several curing agents are available, NR is almost always vulcanized by sulfur-containing curing systems. Despite poor chemical resistance and processability, NR shows good elastic properties, resilience, and damping. The low aging resistance of NR is due to its poor stability towards ozone and oxygen. This rubber is mainly used for the production of tires, gloves, toys, elastic bands, erasers, and sports equipment [10,11,12].

#### 1.2.2. Synthetic Rubbers

##### Styrene-Butadiene Rubber 

SBR is made from the copolymers of styrene and butadiene (Figure 2), but its properties are mainly affected by the polymer chains’ structure and styrene content. SBR cannot crystallize under stress and is mostly vulcanized by sulfur agents. Currently, free radical copolymerisation in emulsion and anionic copolymerisation in solution are the main copolymerisation methods for SBR preparation. SBR has low mechanical strength, making it necessary to add reinforcing fillers into its formulation. SBR has been used in automotive industries, especially for car tires, because of its high abrasion resistance, thermal stability, and resistance against crack formation (better than NR and BR). However, SBR is less chemically reactive with slow curing kinetics, which requires more accelerators [12,13].

##### Nitrile-Butadiene Rubber 

As shown in Figure 3 NBR is made from the copolymers of acrylonitrile and butadiene via radical copolymerisation in emulsion at low temperature (5–30 °C). NBR does not crystallize under stress and has low tensile strength, but shows good resistance to non-polar solvents, fats, oils, and motor fuel. Oil resistance is directly dependent on the acrylonitrile content. The NBR structure is determined by its preparation method and changes from linear to highly branched molecules according to the copolymerisation temperature. Swelling resistance in non-polar agents and Tg both increase with increasing acrylonitrile content. NBR has been widely used for sealing tubes, oil transport equipment, and other devices with oil resistance [12].

##### Ethylene-Propylene-Diene Monomer 

EPDM is a terpolymer of ethylene, propylene, and a non-conjugated diene with residual unsaturation in the side chain. This synthetic rubber with a non-polar backbone shows better resistance to heat, light, and ozone compared to unsaturated rubbers (NR or SBR). One of the most important grades of EPDM is with 5-ethylidene-2-norborene (ENB) as a diene (Figure 4). EPDM’s properties depend on the ethylene and propylene content. The most significant properties of the vulcanized EPDM are the excellent resistance to atmospheric aging, oxygen, and ozone up to 150 °C. EPDM can be cured by peroxide or sulfur systems, and these rubbers are extensively used as sealing materials [12,14,15]. Despite peroxidic curing, sulfur vulcanisation of EPDM shows complex reactions induced by sulfur during crosslinking, and a few kinetic numerical models are available on the accelerated sulfur vulcanisation of EPDM [16,17].

##### Polyurethane 

PU is produced by the polyaddition of diisocyanates and polyols (an alcohol having two or more hydroxyl groups) (Figure 5). PU can be obtained in various chemical structures and different properties because of the types of monomers, composition ratios, and reaction conditions. PU has several advantages such as good abrasion and tear resistance, tensile strength, oxygen and ozone resistance, and a low friction coefficient. The largest application of PU is in automotive industries as dampers, flexible connections, and electric lines [18].

##### Silicone Rubber

Silicon rubber, also known as siloxanes, polyorganosiloxanes, or polysiloxanes, is produced by multilevel hydrolysis and subsequent condensation of dimethyldichlorosilane in an acid medium or by ring opening polymerisation of cyclotetrasiloxane, catalyzed by strong acids or bases. The polymer backbone is based on a chain of silicon and oxygen atoms rather than carbon and hydrogen atoms. Silicone rubbers with a very flexible structure show high stability over a wide range of temperatures (−70 °C to 250 °C) [19]. As shown in Figure 6, there are four primary groups identified by letters forming a typical polysiloxane. Silicon rubbers are also resistant to oxygen and ozone ageing, so this rubber is mainly used for the manufacture of tubing for ozone transport. Finally, silicon rubbers are highly adhesive, hydrophobic, and biocompatible, making this rubber an ideal material for medical implants and other devices biocompatible with human organisms [10,14].

## 2. Recycling

It is well known that polymer decomposition (biodegradation) takes a long time and causes harmful environmental effects. Therefore, polymer wastes’ disposal is a serious environmental issue. Tires containing almost 50% rubber are polymeric materials. The global production of rubber materials in 2017 was about 26.7 Mtons divided into 12.31 Mtons of NR and 14.46 Mtons of synthetic rubber [9]. Discarded rubber pipes, belts, and shoes are various types of waste rubber products. However, the tire industries, as the main application of rubbers (65% of the global production), generate the largest amounts of rubber waste materials. Therefore, rubber recycling is often defined as tire recycling. Currently, 1.5 Bn tires/year are discarded worldwide containing up to 90% of vulcanized rubber that cannot be easily recycled (reprocessed) due to their complex crosslinked structure [9]. Vulcanized rubbers are being used in tires manufacturing since these thermoset materials can sustain severe mechanical and thermal conditions while their properties do not change with temperature. The chemical composition of tires influences their mechanical behavior and lifespan. As shown in Table 1, the typical tire compositions for passenger cars (7.5–9 kg) and trucks (50–80 kg) are different based on the rubber type, as well as the other components [20].

Waste tires are rich materials due to their composition and properties and thus the sources of valuable raw materials. Waste tires can be categorized as worn tires or end of life tires in which some of these worn tires are still suitable for on the road use. However, end of life tires cannot be used for tire manufacture. The incorporation of different additives such as stabilisers, antioxidants, and anti-ozonants into the vulcanized rubber compounds make them resistant to biodegradation, photochemical decomposition, chemical reagents, and thermal degradation. Due to this complex formulation, finding practical methods at a suitable cost for waste tires’ recycling is a serious dilemma for the tire industries. Landfilling is the easiest approach to get rid of waste tires. However, there are several drawbacks. For instance, impermeable discarded tires might keep water for a long period of time and support sites for mosquito larva breeding, which cause deadly diseases such as dengue and malaria [1]. Several works have been reported on recycling end of life tires for energy recovery [23] and pyrolysis [24]. Waste tires, which contain more than 90% organic materials with a heat value of 32.6 MJ/kg (the heat value of coal is 18.6–27.9 MJ/kg), have been used for energy recovery purposes [1]. For example, waste tires are used as a fuel source in cement kilns, which is more environmentally friendly compared to coal combustion. Moreover, waste tires are used as fuel for the production of steam, electrical energy, pulp, paper, lime, and steel. However, burning tires as fuel releases hazardous gases and only recovers 25% of the energy used for the rubber production [25]. Furthermore, the pyrolysis of waste tires decomposes the rubber component to produce carbon black, zinc, sulphur, steels, oils, and gas. However, the high operating costs of the pyrolysis plants limit the wide application of this method [26]. Some environmentally friendly recycling techniques have been developed such as triboelectric separation, froth flotation, and laser-induced breakdown spectroscopy. However, these methods are expensive, and the obtained recycled rubbers vary in cleanliness, size, shape, and surface topography quality [21,27,28]. Although vulcanized waste rubbers are difficult to recycle, they are very durable, strong, and flexible materials, which can be used as ideal fillers in composite production [9].

Therefore, an interesting option is to blend waste tires with plastics (by the action of heat and pressure) to decrease the final costs of the products due to a lower amount of virgin material being used. Waste tires need to be shredded (grinding) into smaller particles (downsizing) for easier incorporation into plastic matrices. Usually, pneumatic separators and electromagnets are used for the separation of textiles and steels from waste tires, respectively [1]. Several methods of waste tire downsizing processes are presented in Table 2, resulting in different surface characteristics and the size of GTR. Cryogenic processes lead to clean granulates without surface oxidation. Shredded tires can be used in virgin/fresh polymers such as rubbers, thermoplastics, and thermoset blends for civil engineering, automotive applications, sports equipment, and others. Blends of rubber with thermoplastics are consuming a large amount of waste tires, as discussed in the next section [1,29].

## 3. Thermoplastic Elastomers 

Thermoplastic resins are being broadly used for melt blending with waste rubber powder to form TPE compounds. TPE are composed of an elastomeric component as a soft fraction and a non-elastomeric material as a hard segment, which is a thermodynamically incompatible system. TPE compounds benefit from the processability of thermoplastics and the properties of glassy/semi-crystalline thermoplastics combined with soft elastomers. TPE compounds can be prepared by extrusion through the dissociation of hard domains at high temperature and shear followed by cooling and solidifying the polymer melt. TPE materials are categorized into thermoplastic olefin (TPO), thermoplastic natural rubber (TPNR), thermoplastic vulcanizate (TPV), thermoplastic polyurethane (TPU), styrene block copolymer (SBC), polyether block amide (PEBA), and copolyester (COPE) [30].

### 3.1. TPE Structure

TPE compounds can be obtained by three different structures and morphologies as:Block copolymers consisting of elastic and non-elastic blocks;Rubber/thermoplastic blends;Dynamically vulcanized rubber/thermoplastic blends.

#### 3.1.1. Block Copolymers

TPE based on block copolymers consist of multi-block copolymers for which the end of these blocks can be crystallized and linked together, forming a crosslinked network. The main fraction of block copolymers is the amorphous phase with rubber-like properties. Several copolymers have been used in this category such as TPU, SBC, PEBA, and COPE. Figure 7 resents a schematic representation of a TPE copolymer illustrating the rigid crystalline segments and rubbery blocks as a continuous domain of soft rubbery chains. Under deformation, the hard blocks remain crystalline and never deform, so TPE deformation is governed by the soft rubber domains. Going through the melt temperature, the copolymer chains start to flow, and the material can be processed like all thermoplastic polymers [31,32].

#### 3.1.2. Rubber/Thermoplastic Blends

Typical TPE compounds are prepared by direct melt blending of an elastomer with a thermoplastic by internal mixing (batch) or extrusion (continuous). TPO is a well known type of TPE based on melt blending of a rubber and a polyolefin such as polypropylene (PP), low-density polyethylene (LDPE), linear low-density polyethylene (LLDPE), and HDPE. As shown in Figure 8, the thermoplastic is the continuous phase, but the morphology of TPO is not fixed as the rubber phase shape and size might change by coalescence or rupture during high shear processing. Since the dispersed rubber phase is not crosslinked with the thermoplastic, TPO can be easily prepared at low cost. TPO have been extensively used in the transportation sector including automotive exteriors and interior fascia [31,32].

TPE compounds are mostly prepared from heat resistant rubbers such as EPDM. NR has been introduced in the TPE production especially after the development of dynamic vulcanisation through phenolic curatives. TPE containing NR as the elastomer component melt blended with thermoplastics are known as TPNR. Usually, TPNR compounds are melt blended via internal mixer or co-rotating twin screw extruders. Several thermoplastics such as polystyrene (PS) [34], polyamide 6 (PA6) [35], ethylene-vinyl acetate (EVA) [36], and poly(methyl methacrylate) (PMMA) [37] are reported to be used in TPNR production. Furthermore, different polyolefins (PP, LDPE, HDPE) have been broadly used for TPNR preparation [38]. For example, melt blending of NR and HDPE results in a combination of the excellent processing properties of HDPE and the elastic properties of NR to produce TPNR for automobile components. Since HDPE and NR are nonpolar materials with totally different melt viscosity and MW, they show poor interfacial adhesion. Not only compatibilisers have been reported to enhance interaction between both phases, but also processing oils have been used for their softening ability (plasticizing), processability improvement (lubrication), and elastic recovery [39,40].

#### 3.1.3. Thermoplastic Vulcanizates 

TPV compounds are based on melt blending of the elastomer with the thermoplastic at high temperature and shear through dynamic vulcanisation or an in situ crosslinking process. The dynamic vulcanisation process crosslinks the elastomer component dispersed in the continuous thermoplastic phase, even if its volume fraction is above 50%. The dispersed particles’ (rubber phase) size directly affects the physical properties of TPV with 1 µm being the optimum rubber particles size (Figure 9) [32].

The preparation of TPV compounds is expensive and requires complex processing since the dispersed rubber phase needs to be crosslinked during mixing. The high amount of rubber (>50 wt.%) with high crosslinking density leads to high elasticity and rubber being the continuous phase, while the uniformly dispersed rubber phase is essential for the desired mechanical properties of TPV. On the other hand, a continuous plastic phase is required for appropriate processability. Altogether, the phase inversion of the rubber phase from a continuous phase (in the premix) to a dispersed phase (in the TPV) shows a dominant role in the preparation of TPV compounds. As shown in Figure 10, a high amount of rubber (50–80 wt.%) is melt blended with the thermoplastic (20–50 wt.%) at high temperature and shear stress. Dynamic vulcanisation is performed after adding the curing agents and other additives into the premixed blends under the same processing conditions to crosslink the rubber phase. Rubber crosslinking and breaking up occur simultaneously, so the phase inversion occurs. Then, intensive mixing is required to achieve uniform dispersion of rubber particles in the thermoplastic matrix. Since the vulcanized rubber domain and thermoplastic matrix show poor interfacial adhesion, compatibilisation is required to achieve TPV with good overall properties and mechanical strength. Compatibilisers can improve the interfacial adhesion by decreasing the surface tension of the TPV components [41,42].

## 4. Compatibility

Melt blending of waste rubber with a thermoplastic resin is an upcycling process and adequate technique for waste tires’ recycling. However, interfacial incompatibility between both phases is a critical issue in melt blending processes. Thermodynamically, due to large unfavorable enthalpy, the incompatibility of polymer blends leads to phase separation, weak interfacial adhesion, and poor mechanical properties. Therefore, controlling the morphology and interfacial tension plays an important role in determining the properties of polymer blends. Miscibility and compatibility in polymer blends are closely related and are often confused since both terms contribute to the morphology and properties. Generally, miscibility results in one phase, while compatibility creates a disperse phase (interphase) for which its size and stability is determined by interfacial interactions [43].

The basic thermodynamic relationship controlling mixtures is:ΔG_m_ = ΔH_m_ − T ΔS_m_(1)
where ΔG_m_ is the free energy of mixing, ΔH_m_ is the enthalpy of mixing, and ΔS_m_ is the entropy of mixing at the temperature T.

The miscibility theory for polymer blends was introduced by Flory and Huggins [44]. Based on this theory, ΔS_m_ is the entropy factor and corresponds to the disorder or randomness value that is always positive; so, it is favorable to mixing or miscibility. In polymer-polymer mixtures, the entropy of mixing has a negligible value, and the enthalpy of mixing (ΔH_m_) is the dominant factor to determine miscibility. ΔG_m_ will be negative if and only if ΔH_m_ is negative: exothermic mixing requiring specific interactions between the components of the blend.

Incorporation of additives is a common method to improve the miscibility of polymer blends by decreasing their interfacial tension, which is called compatibilization. In fact, the main objectives of compatibilization are:Lowering the interfacial tension,Controlling the morphology by size reduction and stabilisation of the dispersed droplets to prevent their coalescence,Increasing the interfacial adhesion between the phases, leading to better stress transfer and mechanical properties [45].

Physical and chemical compatibilisation methods are two main strategies for blend compatibilisation. For example, Iyer and Schiraldi [46] reported that the functional groups of additives (copolymers or nanoparticles) can interact with one or both of the polymers, thereby improving the compatibility of polymer blends.

Physical compatibilisation of polymer blends is based on applying external energy. Generally, the crosslinked structure of the vulcanized rubber is destroyed with energy sources to create physical entanglements and increase the interaction between the thermoplastic and rubber molecules. Physical compatibilisation (mechanical or thermo-mechanical stresses assisted by oil), high energy radiation (microwave or γ radiation), and ultrasonics (ultrasonic waves) are conventional physical compatibilisation methods.

Chemical compatibilisation of polymer blends is conducted through non-reactive and reactive approaches using chemical agents [47]. In non-reactive methods, a block or graft copolymers with chain units similar to the blend components are used. Kumar et al. [48] studied an immiscible blend of GTR/LLDPE and used SBR, NR, and EPDM to improve the compatibility of polymer blends. According to their results, the blends containing EPDM showed the highest mechanical properties (almost 60–70% improvement in tensile strength) as a result of improved interaction and compatibility between the components. Recently, inorganic nanoparticles (NP) have been used as compatibilisers since they can bridge immiscible polymers and offer compatibility.

In reactive compatibilisation, copolymers are generated in situ during the melt blending process. Copolymers’ formation might occur by reaction between the end-groups of the first polymer with the end-groups or pendant groups of a second polymer [45]. Furthermore, dynamic vulcanisation involving the immobilisation of the dispersed phase via crosslinking can also improve the blend compatibility. Usually, the vulcanized rubber as the dispersed phase is a crosslinked component distributed in the continuous thermoplastic phase [42].

### 4.1. Copolymers

Copolymers are extensively used as compatibilisers in immiscible polymer blends, and their efficiency is determined by their composition, chain length, and configuration (Figure 11). Copolymers need to have segments that can interact with each polymer in the blend [45].

For instance, Shanmugharaj et al. [49] used polypropylene grafted maleic anhydride (PP-g-MA) as a compatibiliser in PP/GTR blends by using allylamine grafted GTR and reported 10–20% tensile strength improvement of the PP-g-MA containing compound compared with unmodified blends as a result of enhanced compatibility and interaction between all the components. Furthermore, Kim et al. [50] compatibilized acrylamide (AAm) modified GTR/HDPE blends with PP-g-MA and reported impact strength improvements of the AAm-grafted powder-filled composite compared with those of the unmodified powder-filled system and due to the bonding effect between rubber powders and the compatibiliser (Figure 12). Similar studies also focused on using copolymers as compatibilisers in TPE blends [51,52,53,54].

### 4.2. Nanoparticles (NP)

More recently, inorganic NP with a large specific surface area and high aspect ratio such as graphene (specific surface area 2600 m^2^/g and aspect ratio 200–1000) [55], single walled CNT (specific surface area 1315 m^2^/g and aspect ratio >1000) [56], and nanoclay (natural montmorillonite clay specific surface area 750 m^2^/g and aspect ratio 200–1000) [57] have been used as compatibilisers in polymer melt blending in addition to their application for improving the mechanical, thermal, and barrier properties [58,59,60]. The Flory–Huggins thermodynamics theory of mixing clarifies the phase separation in a ternary system containing two polymers and NP. However, droplet stabilisation against coalescence is not clearly understood. There are different mechanisms for the NP compatibilisation effect in polymer blends. Based on thermodynamics compatibility, the large specific surface area and high aspect ratio of inorganic NP adsorb the polymer chains on their surface to increase the stabilizing energy leading to the negative overall free energy of mixing and a thermodynamically compatible system. On the other hand, kinetics compatibility is related to the selective localisation of the NP at the polymers interface by decreasing the interfacial tension and preventing droplet coalescence during melt blending. The compatibilisation efficiency of NP is affected by their migration and localisation in phases during melt blending, which can be determined by processing parameters (compounding sequence, melt compounding time, and shear rate) [61]. Moreover, blend morphology depends on the viscosity ratio and the interfacial tension between the polymer phases. For instance, a finer morphology is achieved in polymer blends as the viscosity ratio between the matrix and dispersed phases is closer to one [62], as well as low interfacial tension between the blend components [63]. NP are recognized as appropriate compatibilisers to decrease the interfacial tension of polymer blends and stabilize the morphology depending on their localisation. If the nanofillers migrate to one phase of the co-continuous blend, they form a percolated particle network in one phase and prevent coarsening related to the increased viscosity [64]. On the other hand, selective localisation of NP at the interface of polymer blends can stabilize the co-continuous structure. NP jammed at the interface are more effective than percolated particle networks within one of the two phases by suppressing the coarsening phenomena [65]. NP localisation can be predicted by measuring its wetting coefficients (ω) defined as:(2)$$\omega =\left[\left(\gamma_{NP/x}-\gamma_{NP/y}\right)/\gamma_{x/y}\right]$$
where $\gamma_{NP/x}$, $\gamma_{NP/y}$, and $\gamma_{x/y}$ are the interfacial energies (or interfacial tensions) between NP–polymer (*x*), NP–polymer (*y*), and polymer (*x*)–polymer (*y*), respectively. All these interfacial energies can be theoretically calculated based on the Owens–Wendt equation [66] by measuring the dispersive ($\gamma^{d}$) and polar ($\gamma^{p}$) part of the surface energies:(3)γxy=γx+γy−2[(γxdγyd)12+(γxpγyp)12]

Based on Equation (2), if the wetting coefficient is higher than one (ω > 1), the NP thermodynamically prefer to stay in the polymer (*y*), while NP locate in the polymer (*x*) when ω < −1. Ideally, NP migrate to the interface between both phases when −1 < ω < 1 (Figure 13b) and act as smart/functional barriers inhibiting droplets’ coalescence [58,61].

Several inorganic NP have been used for both reinforcing and compatibilisation effects in immiscible polymer blends. However, the main challenge in using NP for blend compatibilisation is their poor dispersion in the polymer matrix due to particle agglomeration, limiting their efficiency [45].

## 5. Rubber Modification

Several methods, such as graft polymerisation, radiation-induced modification, and gas modification, have been proposed to modify rubbers. Currently, rubber surface modification techniques have been performed at the laboratory scale. The purpose of rubber modification is to introduce oxygen functional groups (peroxy, hydroperoxy, hydroxyl, and carbonyl) on the rubber surface to interact with polar polymers or reactive compatibilisers to improve the interfacial adhesion between the polymer and rubber. Conventional oxidizing agents including potassium permanganate (KMnO_4_) [67], nitric acid (HNO_3_) and hydrogen peroxide (H_2_O_2_) [68], and sulphuric acid (H_2_SO_4_) [69] have been used. Moreover, grafting monomers onto rubber particles through free-radical initiation or photo-initiation can prevent particles’ agglomeration, leading to smaller particle size and more homogeneous distribution within the continuous polymer matrix to achieve better blend properties [1,27,70].

### Reclamation and Devulcanisation

Vulcanized rubbers are infusible and insoluble materials with a 3D crosslinked structure (100% gel content), which are difficult to process and reprocess for further compound production. Therefore, these rubbers need to be partially soluble with lower crosslink density, which can be achieved by partially destroying the initial crosslinked structure, giving more chain mobility (molecular freedom). The soluble fraction can interact and bond with the polymer matrix chains. Thermomechanical, thermochemical, ultrasonic, and microwave are common techniques for partial breakup of the crosslinked structure of vulcanized rubbers. Regardless of the method used, there are two concepts related to the process of destroying the crosslinked structure of rubber including devulcanisation and reclamation. Reclamation is based on the scission of C–C bonds in the rubber backbone to reduce the MW and obtain some plasticity. On the other hand, devulcanisation is the specific cleavage of S–S and C–S bonds, partially destroying the 3D network to produce plasticity. In an ideal devulcanisation process, the rubber backbone should not be damaged. However, selective breakup of the crosslinked structure inside vulcanized rubber is not possible without damaging some C–C bonds in the backbone. Table 3 reports the energy required for breaking the different bonds of crosslinked rubbers. In general, reclamation and devulcanisation might occur at the same time, making their differentiation difficult in a specific process (Figure 14) [1].

## 6. TPE Compatibilisation

Vulcanized and reclaimed rubber are exposed to severe conditions (shear stress, thermal and chemical degradation, radiation) in their lifetime and recycling processes, so the properties of the resulting TPE differ from compounds based on virgin materials. Furthermore, rubbers contain several fillers that might limit possible improvement of blend properties. TPE compounds based on polyolefin (especially PE) have received a great deal of attention because they are easy to process, and the materials are easily available at low costs. However, the performance of these blends depends on the nature and concentration of each component, as well as their interaction. The compounds need to show at least 100% elongation at break and compression set lower than 50% to be recognized as good TPE materials [72]. It is known that polymer blend properties significantly depend on the interfacial adhesion between both phases and the size of the dispersed phase inside the continuous matrix. Poor interfacial adhesion between the rubber and thermoplastic phases leads to low mechanical properties. In fact, the vulcanized rubber molecules do not have enough freedom to entangle with the thermoplastic molecules to create strong bonding. Therefore, the interfacial adhesion and morphological behavior of TPE blends are important parameters to control/optimize the composition and processing conditions for high performance compounds [1,70].

### 6.1. Effect of Rubber Particles’ Size and Loading

Considering the size of the dispersed phase, small rubber particles usually show better mechanical properties than larger particles due to a lower probability of failure/crack formation. Ismail et al. [73] studied the effect of three different GTR sizes (250–500 μm, 500–710 μm, and 710 μm–1 mm) on the mechanical properties of PP/GTR blends. They reported that blends containing smaller GTR particles (250–500 μm) showed higher equilibrium torque due to high friction associated with the higher surface area of the smaller GTR particles. As shown in Figure 15, the blends containing small GTR particles also showed the highest elongation at break (20%). However, the values were low because of the crosslinked structure of the GTR particles and poor adhesion with the PP matrix, resulting in easy crack initiation and rapid crack propagation.

Sonnier et al. [67] used three different rubber particle sizes (380–1200 µm) in GTR/LDPE compounds. They did not achieve a significant difference in the mechanical properties of GTR/LDPE (50/50 wt.%) blends (impact energy ~2.6 kJ/m^2^ for all blends with different rubber particle sizes). Therefore, they suggested that controlling the GTR particles’ size is not the only parameter to achieve significant mechanical properties’ improvement. It has been reported that the effective rubber particle size to improve the mechanical properties of TPE is around 500 µm or less (Figure 16). However, at high rubber concentration (above 50 wt.%), the effect of rubber particle size is less important since low interfacial adhesion is the dominant parameter controlling the mechanical properties. In fact, substantial drops in the tensile strength and impact strength of TPE compounds filled with vulcanized rubbers are related to low interfacial adhesion, rubber particles’ agglomeration, and void formation at the interface between the rubber and thermoplastic phases. Due to a mismatch in polarity, melt viscosity, and MW of both materials, the interfacial adhesion is weak. Poor interface quality leads to high interfacial tension and GTR particle agglomeration, facilitating voids’ formation around the rubber particles. As shown in Figure 17, increasing the rubber concentration resulted in the formation of more defects and cracks in GTR/EVA compounds. A clear indication of low interfacial adhesion is confirmed by the clean and easy removal of rubber particles (pull-outs) from the EVA matrix [74].

### 6.2. Non-Reactive Compatibilisation

It is also possible to improve the interfacial adhesion of immiscible polymers blends via compatibilisation methods. The addition of compatibilising aids (copolymers or nanoparticles), surface modification of the materials, as well as a variety of devulcanisation methods and processing aids (solvents) are conventional techniques to enhance the compatibility of TPE blends. Incorporation of block or graft copolymers into polymer blends decreases the interfacial tension and promotes interaction between polymers. Different compatibilisers such ethylene-acrylic acid copolymer (EAA) [75], chlorinated polyethylene (CPE) [76], PE-g-MA [77], ethylene-co-glycidyl methacrylate copolymer (E-GMA) [78], epoxydised NR [79], styrene-butadiene-styrene block copolymer (SBS) [80], and EVA [30] have been used in TPE compounds. For example, PE-g-MA showed good efficiency for improving the mechanical properties of TPO compounds as a result of a reaction between the anhydride groups grafted onto polyethylene (PE) with hydroxyl groups/unsaturated bonds on the GTR particles’ surface. Therefore, using PE-g-MA as a compatibiliser can reduce the interfacial tension, improve the dispersed phase uniformity, decrease the domain size, and maintain the blends’ morphology stability [81]. Esmizadeh et al. [77] studied the effect of reactive compatibilisation on the mechanical and morphological properties of TPV blends containing HDPE/reclaimed rubber (RR). They used PE-g-MA and peroxide as the compatibiliser and vulcanizing agent, respectively. Analysis of the torque values showed increasing trends of the plateau region (equilibrium value) with increasing RR content due to the restricted chain mobility and difficult dispersion of crosslinked rubber particles in HDPE. A similar observation was reported by Ismail et al. [73] in which the stabilized torque increased from 4 Nm to 8 Nm with increasing GTR content from 20 wt.% to 60 wt.% due to a good dispersion of hard crosslinked rubber particles in PP (Figure 18).

Furthermore, reactive compatibilisation and dynamic vulcanisation can increase the torque plateau due to the increased viscosity of the system [77]. Generally, TPE compounds show a shear-thinning (pseudo-plastic) behavior, and their viscosity decreases with increasing shear rate. Sae-Oui et al. [82] reported the pseudo-plastic behaviour of NR/HDPE compounds since the complex viscosity decreased with increasing angular frequency (Figure 19). Obviously, increasing complex viscosity was directly related to the NR concentration (complex viscosity (NR/HDPE): 90/10 > 80/20 > 70/30 > 60/40) since the fully NR vulcanized structure restricted flowability.

Moreover, the storage modulus (G’) as a function of angular frequency increased because of less time available for molecular relaxation (Figure 20). Furthermore, G’ increased more at higher NR content (NR/HDPE = 90/10) because of the crosslinked and highly elastic NR content, which gave rise to a stronger elastic response (slope reduction in Figure 20).

Table 4 compares the mechanical strength of compatibilized compounds showing that compatibiliser addition led to higher interfacial adhesion between RR and HDPE. It is clear that a very small amount of vulcanizing agent (0.2 wt.%) is more effective than compatibilisers to improve the mechanical properties. Figure 21 shows that Figure 21. Hardness of high-density *polyethylene* (HDPE) decreased with increasing RR content, which was attributed to the higher concentration of the elastomeric component in the TPE. Furthermore, increased hardness of the compatibilized and dynamically vulcanized blends were related to better interaction between the materials induced by the compatibiliser and the formation of a stronger crosslinked structure (increased rigidity), respectively [77].

Kakroodi et al. [83] used PE-g-MA as a matrix to produce TPE compounds filled with high GTR contents (50–90 wt.%) and compared their mechanical strength with HDPE/GTR compounds. The results showed that TPE containing 50–70 wt.% of GTR in PE-g-MA had very good elongation at break (ε_b_ = 465%) and tensile strength (σ_y_ = 32.7 MPa) at 50 wt.% GTR, while these properties decreased with increasing GTR content to 90 wt.% (ε_b_ = 219% and σ_y_ = 4.6 MPa). Furthermore, the tensile properties of HDPE/GTR compounds, with and without PE-g-MA as a coupling agent, were significantly lower than for the blends with PE-g-MA as the matrix. Therefore, PE-g-MA was shown to be a good matrix to produce TPE with high tensile properties.

Wang et al. [84] worked on the production of TPE compounds based on recycled PE (R-PE)/GTR and investigated the effect of Engage 8180 and Vestenamer 8012 copolymers on the morphological and mechanical properties. They reported better compatibilising efficiency of Engage 8180 on R-PE/GTR compounds than Vestenamer 8012. This behavior was attributed to the interaction and entanglement of R-PE and GTR molecular chains due to the compatibilising effect of the ethylene-octene copolymer (main component of Engage 8180). In fact, the ethylene part was compatible with R-PE, while the octene segment showed entanglement with SBR (main part of GTR). Even though, they reported improved elongation at break of compounds with 10 wt.% Engage 8180 up to 76%. However, the values were still lower than 100%, which implies the need to do more research on the compatibilisation of highly filled TPE compounds, especially when recycled thermoplastic resins are used as the matrix.

### 6.3. Reactive Compatibilisation

In general, better interaction between the components leads to the reduction of the dispersed phase particle size and compatibility improvement. In reactive blending, block or graft copolymers as compatibilisers are formed in situ during mixing. These compatibilisers improve bonding through covalent reactions between the functionalized components in polymers. Grafting through melt blending can be done by a two-roll mill, internal mixer, and twin screw extruder. Kim et al. [50] worked on the surface modification of GTR via grafting of AAm and melt blending of surface modified GTR with HDPE. They used PP-g-MA to induce the reaction between maleic anhydride (MA) and surface modified GTR powders to increase the compatibility between the phases. Both blends containing AAm-grafted GTR and unmodified GTR showed decreasing tensile stress and tensile strain with increasing rubber content. However, the HDPE/AAm-GTR systems showed higher tensile stress and tensile strain. The AAm-GTR filled blends containing 10 wt.% and 20 wt.% rubber did not break and elongated up to 300% and 400%, respectively (Figure 22).

Furthermore, Patel et al. [85] studied the reactive blending of LDPE/NR and LDPE/NBR using acrylic acid (AA) and MA. The reaction mechanism for LDPE/NR modified with MA is presented in Figure 23, while similar reactions are expected for AA-grafted LDPE/NBR. Dicumyl peroxide (DCP) was used as an initiator to generate the free radical sites on the LDPE chains.

The hardness results showed that ungrafted compounds had lower values (Table 5). In fact, the grafting method increased the hardness of both NR and NBR compounds up to 98 Shore A for LDPE/NBR (80/20) blends modified with MA-grafted particles. Moreover, based on tensile results, the strength decreased after aging (air oven at 70 °C for 16 h) due to the degradation of the elastomer phase. After aging, the ungrafted compounds showed a 20% reduction in tensile strength, while the grafted compounds showed only 1–5% reduction in tensile strength, which indicated their higher stability due to grafting [85].

### 6.4. Effect of NP Incorporation

In recent years, anisotropic nanofillers such as nanoclays, CNT, and graphene, with large specific surface area and high aspect ratio, have been used to modify the interfacial adhesion of immiscible polymer blends [58,59]. Incorporation of small amount of NP leads to strong interfacial interaction between the components, improving the mechanical strength and thermal stability of TPE nanocomposites.

Mehta et al. [86] studied the effect of nanoclays on the morphology of PP/EPDM (70/30) blends. They showed important size reduction of the dispersed phase by increasing the nanoclays’ concertation. Generally, the final morphology was influenced by the filler distribution, the viscosity ratio between the components, and the affinity of the filler toward the polymers. Naderi et al. [87] studied the effect of the matrix viscosity and NP content on the mechanical properties and morphology of PP/EPDM/nanoclay compounds. XRD analysis was used to study clays’ exfoliation into nanolayers in the polymer blends. As shown in Figure 24, the addition of 3 wt.% nanoclays (Cloisite 15A) increased the interlayer spacing from 30.44 Å for PP/EPDM (80/20) to 34.62 Å for TPE nanocomposite (PP/EPDM/nanoclays). This behavior was attributed to the intercalation of polymer chains inside the silicate layers. They used a fixed NP concentration since increasing its concertation led to difficult penetration of the polymer chains through the silicate layers and decreased interlayer spacing of the nanoclays.

Furthermore, they reported the effect of the viscosity ratio on the size of the rubber domain in PP/EPDM (60/40) blends. The results showed that the rubber droplet sizes decreased with increasing the viscosity of the PP phase. As mentioned before, a fine dispersion is achieved when the viscosity ratio of the plastic/rubber is close to one. Furthermore, they showed the effect of nanoclays on breaking up the rubber droplets. Increasing the rubber concertation increased the dispersed rubber phase size in the compounds without nanoclays (Figure 25a,c), which indicated the effect of NP on preventing coalescence and reducing the dispersed phase sizes. Therefore, the distribution and domain sizes of the dispersed phase were significantly influenced by the presence of NP and the viscosity ratio between both polymers [87].

In another study, Lopattananon et al. [88] investigated the effect of sodium montmorillonite (Na-MMT) concentration on the mechanical and morphological properties of TPV based on NR/PP (60/40). According to the results, the two phase-separated morphology of the blends changed to a droplet-like structure upon addition of 2–5 phr (parts per hundred resin) nanoclays as a result of a droplet break-up effect.

### 6.5. GTR Surface Modification and Devulcanisation

Another compatibilisation technique is the surface modification of GTR particles via oxidation to improve the interaction between the components. Colom et al. [89] used various acids such as H_2_SO_4_, HNO_3_, and perchloric acid (HClO_4_) for the surface treatment of GTR for melt blending with HDPE. They reported improved rubber interaction with HDPE and higher stiffness for the TPE compounds as a result of rubber rigidification after the acid treatment. As shown in Figure 26, the smooth surface of HClO_4_ treated particles (b) is similar to the surface of untreated particles (Figure 26), which is evidence of poor adhesion. However, HNO_3_ and H_2_SO_4_ (Figure 26c,d) provided a rough surface with several micro-pores and cavities that enhanced the interfacial contact area and interaction between the rubber and the thermoplastic matrix. The micro-roughness topography was related to the acid treatment with sulphuric acid, which led to a decrease in the number of double bonds in the tire chemical structure due to the degradation process of polybutadiene and other unsaturated hydrocarbon polymer chains (diene) on the GTR surface.

In another work, Liu et al. [90] investigated the oxidation of EPDM powder by KMnO_4_ to generate hydroxyl groups by breaking unsaturated C=C bonds in the rubber. The addition of surface modified EPDM into PP containing a small amount of MA grafted chains showed significant elongation at break improvement. It was explained that polar groups on the EPDM surface reacted with MA to form covalent bonds, improving the interaction between the rubber and matrix. Several methods are known to improve the polarity (oxygen concentration) of the rubber surface such as high-energy techniques like plasma, corona discharge, and electron beam [27]. Sonnier et al. [67] studied the production of compatible GTR/HDPE compounds using surface treated rubber particles. They used KMnO_4_ as a common oxidizing agent and γ irradiation for which the energy can induce macromolecular chain scission and free-radical formation, having the possibility to react with the oxygen in air and create polar groups. However, the surface oxidation of GTR was not efficient enough to improve the mechanical properties of HDPE/PE-g-MA/GTR compounds (elongation at break ~24% for all blends with different modified rubber particle sizes).

It should be mentioned that the polarity of an elastomer can influence the interfacial adhesion between GTR and the thermoplastic matrix. Li et al. [30] studied HDPE/GTR blends using EVA and ethylene-octene copolymer (POE) as polar and non-polar compatibilisers, respectively. According to the results, the impact strength and elongation at break of the HDPE/GTR/POE (60/20/20) compounds were 417 J/m and 129%, both of which were higher than 175 J/m and 82% for the HDPE/GTR/EVA (60/20/20) compounds. After morphological analysis, this behavior was explained by better homogeneity and encapsulation of the GTR particles by non-polar copolymers, helping the thermoplastic matrix to deform under applied forces. Furthermore, Formela et al. [80] investigated the effects of non-polar elastomer (partially crosslinked butyl rubber and SBS block copolymers) on the morphological and mechanical properties of LDPE/GTR blends. GTR particles were encapsulated by the elastomer phase, which was compatible with the thermoplastic phase, improving the interfacial adhesion with the LDPE matrix. As shown in Figure 27, small GTR particles showed higher interfacial adhesion as a result of better encapsulation of the GTR particles by the elastomer. Moreover, the compounds containing SBS (branched Kraton 1184) showed up to 125% elongation at break, which is twice the value of LDPE/GTR (50/50), indicating better compatibility with both LDPE and GTR.

Reclamation and devulcanisation of rubber have also been used to improve the compatibility and processability of TPE. In fact, destroying the crosslinked rubber structure, as well as co-crosslinking at the interface enhanced polymer chains mobility and the mechanical strength of the resulting compounds. Furthermore, the presence of RR short chains and processing oil enhanced the processability and elongation at break of RR containing compounds due to a plasticisation effect. The crosslinked gel part of recycled rubber particles acts as stress concentration points, so increasing the rubber concentration (gel content) in the blends leads to increased crosslink density, producing lower tensile strength and elongation at break [1]. Sripornsawat et al. [91] studied the devulcanisation reaction time and temperature through a relation between the soluble fraction (sol) and crosslink density. According to their results, it was required to perform the devulcanisation process in a short time to prevent recombination of free-radicals to form new covalent bonds. Furthermore, increasing the reaction time leads to the generation of more reactive radicals forming new links and increasing the crosslink density. As shown in Figure 28, the optimum devulcanisation time is 4 min to obtain the maximum tensile strength (3.7 MPa) and elongation at break (57%) for these samples. The authors also investigated the effect of devulcanisation on the mechanical and morphological properties of TPV based on blends of COPE with devulcanized rubber (DR) and undevulcanised rubber (UDR). It was expected that DR had more unsaturated and uncrosslinked chains to participate in dynamic vulcanisation. As shown in Figure 29, the polar functional groups on the surface of DR domains interacted with the COPE matrix and showed better interfacial adhesion. High interfacial adhesion induced compatibility between the components, leading to improved tensile strength and elongation at break of TPV containing DR.

Crosslinked polymers do not dissolve in solvents due to their chemically bonded hydrocarbon chains. However, these links do not prevent the swelling of crosslinked polymers. Polymer swelling is defined as a volume increase of the gel fraction by a liquid or a gas [91]. Macsiniuc et al. [92] proposed a pre-treatment of rubber particles in a solvent to improve the compatibility between rubber and plastic for TPE preparation. This method is based on swelling the rubber chains by a solvent, improving the penetration of the dissolved thermoplastic matrix molecules into the crosslinked rubber network. Macsiniuc et al. [92] studied the swelling behavior of SBR particles and the penetration of matrix molecules in the crosslinked rubber structure. They reported increased Young’s modulus (468 to 652 MPa), tensile strength (5.14 to 9.39 MPa), and impact strength (35.2 to 50.1 J/m). These mechanical properties’ improvement was attributed to the effect of swelling SBR particles in tetrahydrofuran (THF), which allowed the PS molecules to enter the pores/voids. Consequently, interfacial adhesion of PS/SBR was enhanced by chain entanglement with a PS matrix. However, immersion time and solvent efficiency affected the swelling and penetration. Similarly, Veilleux and Rodrigue [93] investigated the properties of compounds based on virgin PS with recycled SBR powders (0–94% wt.) using a pre-treatment in solution (toluene) to improve the compatibility between the phases. According to the results of extraction tests and thermogravimetric analysis (TGA), the solution treatment allowed inserting about 7.5% wt. of virgin PS inside the SBR particles, which was lower than the value obtained in a similar work on recycled PS (10.5% wt.) [94]. This difference was attributed to the lower MW of recycled PS, favoring its diffusion into the solvent and the swollen rubber particles. As expected, incorporation of more elastomeric particles (up to 62% SBR) into PS decreased the hardness to 76 Shore A (six units lower than neat PS). Furthermore, the addition of SBR into the rigid PS matrix decreased both the rigidity (modulus) and strength (stress) of the compounds. Furthermore, the incorporation of 62% SBR into PS led to higher impact strength (up to 38 J/m form 22 J/m) due to the presence of more elastomer in the compounds to absorb the impact energy [94].

## 7. Conclusions

Disposal of waste plastic and rubber is a significant issue from an environmental point of view since the natural degradation of these materials takes several years. Vulcanized rubbers are extensively used in a wide range of applications (mainly the tire industries) because of their mechanical strength, excellent durability, abrasion resistance, and low cost.

The recycling of discarded tires as the main fraction of waste rubbers has attracted increasing attention due to the large amounts of waste tires as an environmental issue. However, the complex crosslinked structure and the presence of various additives in the tire composition make their (re)processing difficult. In fact, vulcanized rubber (crosslinked structure) cannot be melted, making tire recycling very difficult. Therefore, it is required to develop technologically possible and cost-effective methods for recycling the waste rubber from scrap tires.

The most straightforward and environmentally friendly method is shredding/grinding waste tires into ground tire rubber (GTR) and using the material (different particle sizes) as fillers in thermosets, virgin rubbers, or thermoplastics (especially recycled resins) to produce thermoplastic elastomer (TPE) compounds. The most convenient size of rubber particles for blending with thermoplastic resins is less than 500 μm since smaller rubber particles are more efficient for improving the TPE mechanical strength. TPE have the combined mechanical properties of thermoplastic/elastomer and easy processability of thermoplastics. Styrene-butadiene rubber (SBR), ethylene propylene diene monomer (EPDM), and natural rubber (NR) are the most used recycled rubber particles for melt blending with thermoplastics to prepare TPE materials. Melt blending of waste rubber particles with recycled plastic is an environmentally friendly and sustainable approach not only for higher consumption of waste polymers, but also because of more economical/eco-friendly advantages. However, low compatibility and weak interfacial adhesion between the rubbers and thermoplastics leads to the low mechanical properties of TPE. Poor interfacial adhesion between the rubber and thermoplastic is more dominant at higher rubber concentration (above 50 wt.%), which significantly deteriorates the mechanical properties of the blends (especially elongation at break and toughness). Therefore, modification techniques are required to obtain recycle-based TPE compounds with appropriate properties. Several compatibilisation methods such as non-reactive and reactive approaches using chemical agents (copolymer/nanoparticles, NP), as well as rubber surface modification through oxidizing agents or reclamation/devulcanisation process and radiation-induced modification were presented here. There is also the possibility of solution treatment using more environmentally friendly (green) solvents. The main objective of these modification techniques is to improve the interfacial adhesion between the rubber particles and thermoplastic matrices to achieve TPE compounds with appropriate mechanical and morphological properties.

It is expected that in the near future, industrial and academic research will focus on the development of green and cost-effective TPE compounds based on recycled polymers. The production of TPE from recycled materials reduces the negative effects of these waste materials’ disposal. It also leads to the production of materials with lower costs. The TPE market is expected to grow significantly in the near future due to increased demand for green and low-cost compounds obtained from waste polymers. Even though the incorporation of recycled rubber (NR and SBR) into thermoplastics has been widely studied, more studies should be done using different types of rubber such as EPDM since this rubber is widely recycled due to its high cost. However, due to variability in the composition of polymer wastes and difficult conditions during their service life, the performance of recycled compounds varies compared to virgin compounds, which needs to be improved. Thermoplastic elastomers seem to be one of the most promising fields of study, and several research works have been conducted on the mechanical and morphological properties of TPE compounds. However, the lack of literature about the thermal, dynamic mechanical, and aging behavior of these compounds highlights the need for more research on TPE preparation and their characterisation.

## Figures and Tables

**Figure 1 materials-13-00782-f001:**
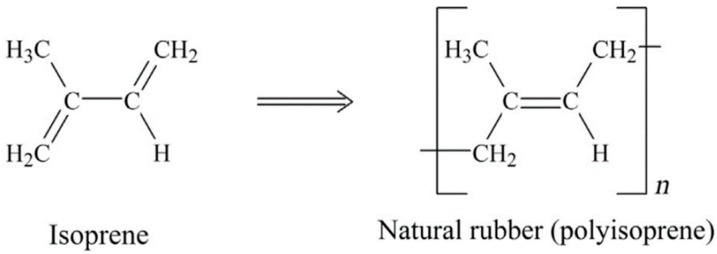
Chemical structure of isoprene and natural rubber (NR) (polyisoprene). Adapted with permission from [12]; copyright 2019 Elsevier Ltd.

**Figure 2 materials-13-00782-f002:**
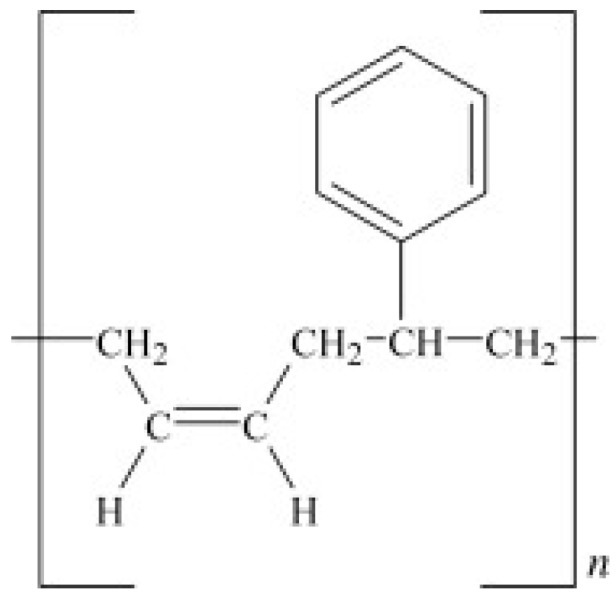
Chemical structure of styrene-butadiene rubber (SBR). Adapted with permission from [12]; copyright 2019 Elsevier Ltd.

**Figure 3 materials-13-00782-f003:**
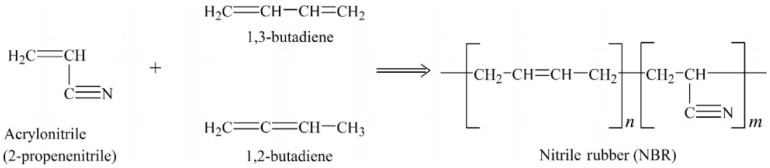
Monomers and polymer structure of nitrile-butadiene rubber (NBR). Adapted with permission from [12]; copyright 2019 Elsevier Ltd.

**Figure 4 materials-13-00782-f004:**
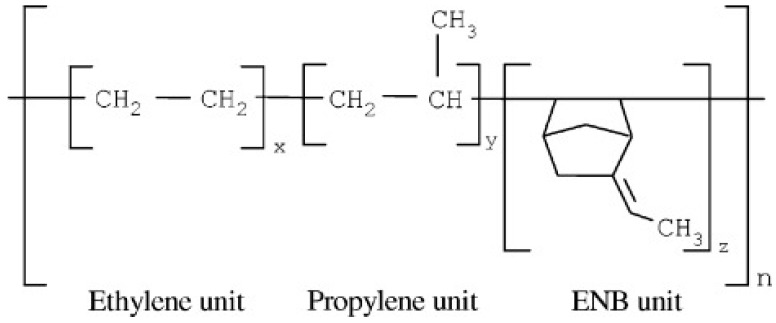
Chemical structure of ethylene-propylene-diene monomer (EPDM) containing 5-ethylidene-2-norborene (ENB) as a diene. Adapted with permission from [15]; copyright 2019 John Wiley and Sons Ltd.

**Figure 5 materials-13-00782-f005:**
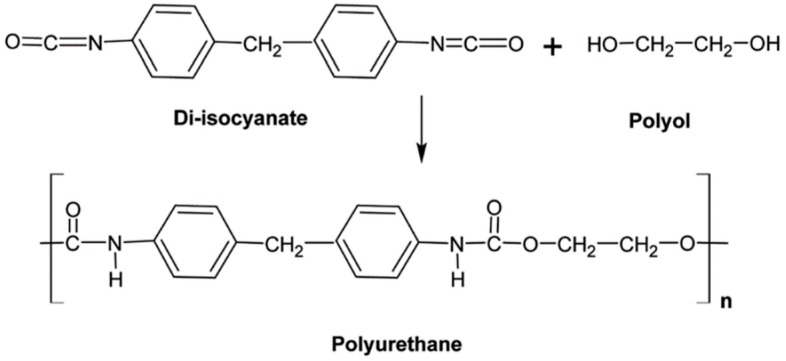
Schematic representation of polyurethane (PU) and its monomers. Adapted with permission from [18]; copyright 2019 RSC Publishing.

**Figure 6 materials-13-00782-f006:**
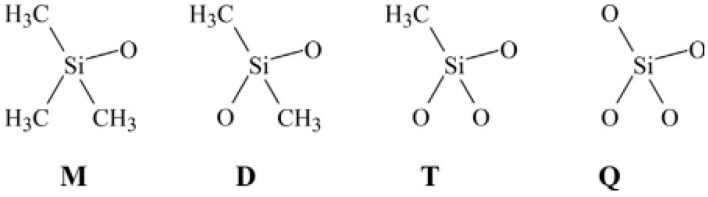
The four groups making polysiloxanes: “M” is trimethylsiloxychlorosilanes (Me_3_SiO), “D” Me_2_SiO_2_, “T” MeSiO_3_, and “Q” silicate (SiO_4_). For “P”, replace Me by phenyl side groups, while for “V”, replace Me by vinyl side groups. Adapted with permission from [12]; copyright 2019 Elsevier Ltd.

**Figure 7 materials-13-00782-f007:**
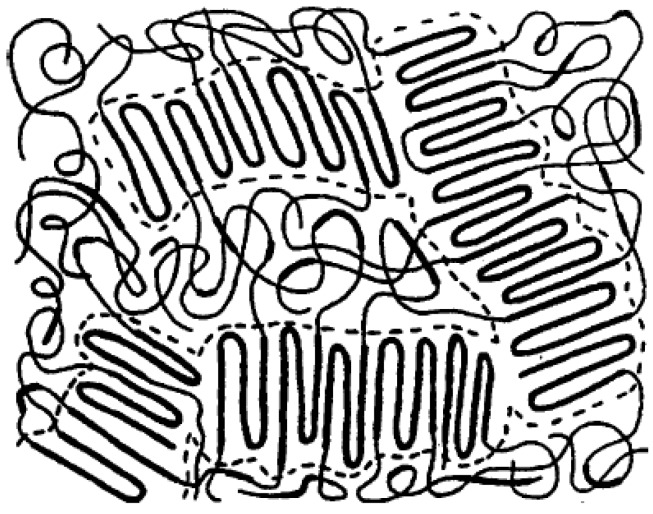
Morphology of a block copolymer thermoplastic elastomer (TPE). Adapted with permission from [33]; copyright 2020 Elsevier Ltd.

**Figure 8 materials-13-00782-f008:**
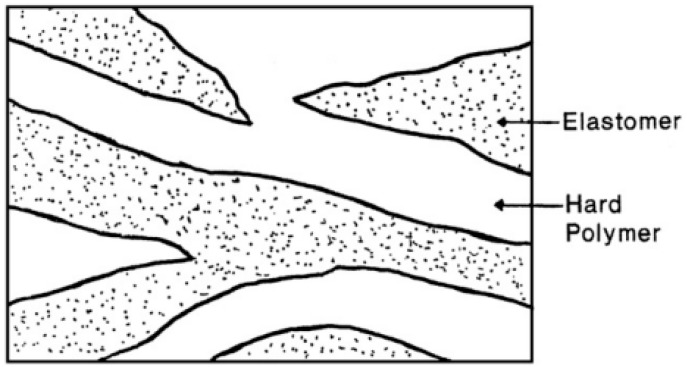
Morphology of rubber/plastic blend thermoplastic elastomer (TPE). Adapted with permission from [32]; copyright 2020 Elsevier Ltd.

**Figure 9 materials-13-00782-f009:**
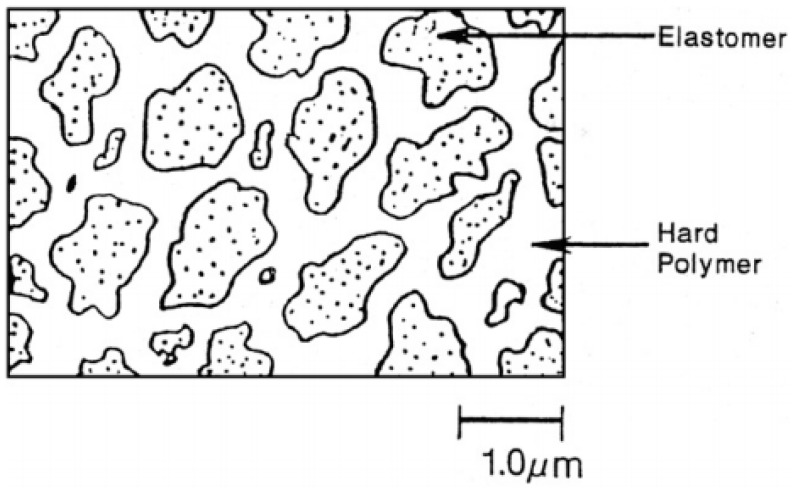
Thermoplastic vulcanizates (TPV) morphology with continuous plastic phase and discrete rubber particles. Adapted with permission from [32]; copyright 2020 Elsevier Ltd.

**Figure 10 materials-13-00782-f010:**
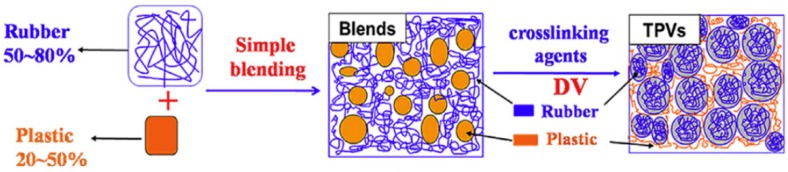
Processing steps to produce thermoplastic vulcanizates (TPV) compounds. Adapted with permission from [42]; copyright 2019 Elsevier Ltd.

**Figure 11 materials-13-00782-f011:**
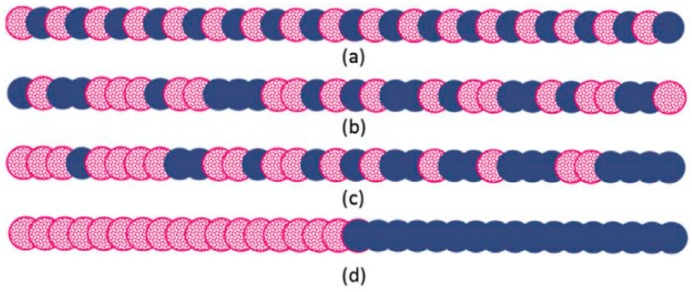
Different structures of linear copolymers: (**a**) alternating, (**b**) random, (**c**) gradient, and (**d**) block copolymers. Adapted with permission from [45]; copyright 2019 Elsevier Ltd.

**Figure 12 materials-13-00782-f012:**
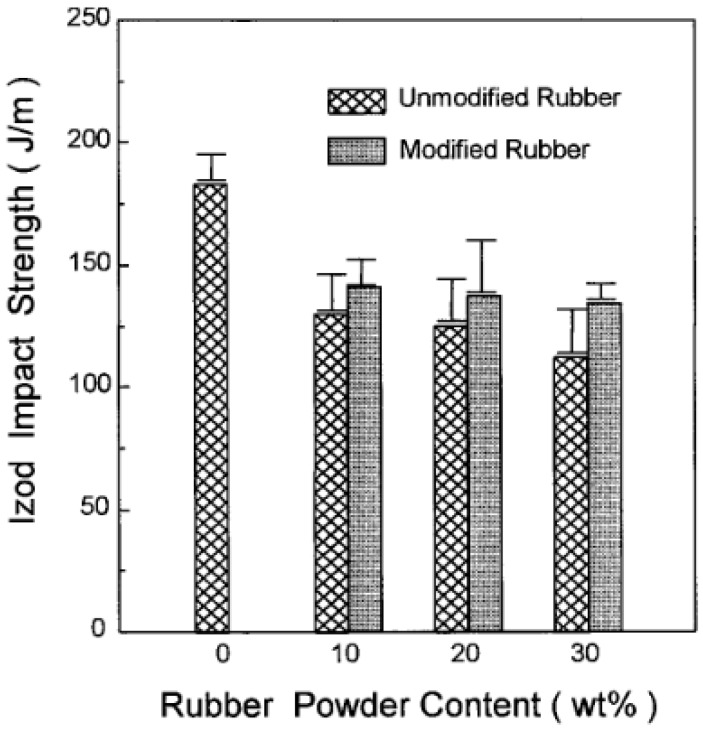
Notched Izod impact strength of high-density polyethylene (HDPE)/ground tire rubber (GTR) composites as a function of rubber content. Adapted with permission from [50]; copyright 2019 John Wiley and Sons Ltd.

**Figure 13 materials-13-00782-f013:**
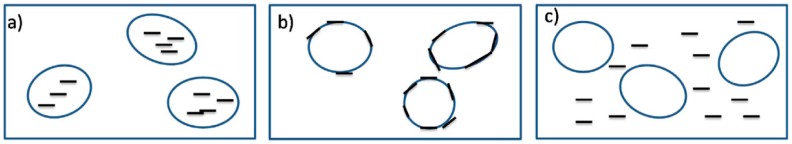
The three possible cases for nanoparticles’ (NP) localisation in an immiscible binary polymer blend: (**a**) in the dispersed phase, (**b**) at the interface (ideal case), or (**c**) in the continuous phase. Adapted with permission from [58]; copyright 2019 Elsevier Ltd.

**Figure 14 materials-13-00782-f014:**
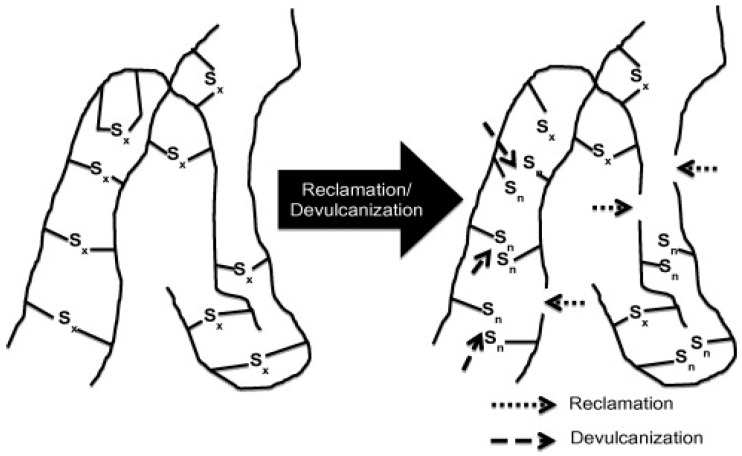
Schematic representation of the devulcanisation and reclamation process. Adapted with permission from [1]; copyright 2020 Elsevier Ltd.

**Figure 15 materials-13-00782-f015:**
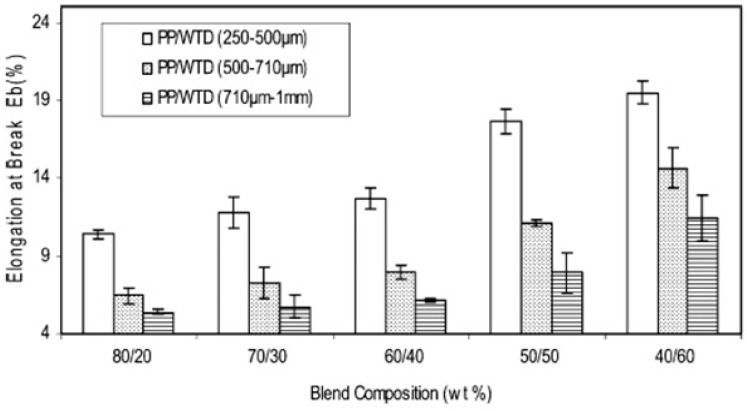
Elongation at break as a function of composition for ground tire rubber (GTR)/polypropylene (PP) blends. Adapted with permission from [73]; copyright 2019 Taylor & Francis Ltd.

**Figure 16 materials-13-00782-f016:**
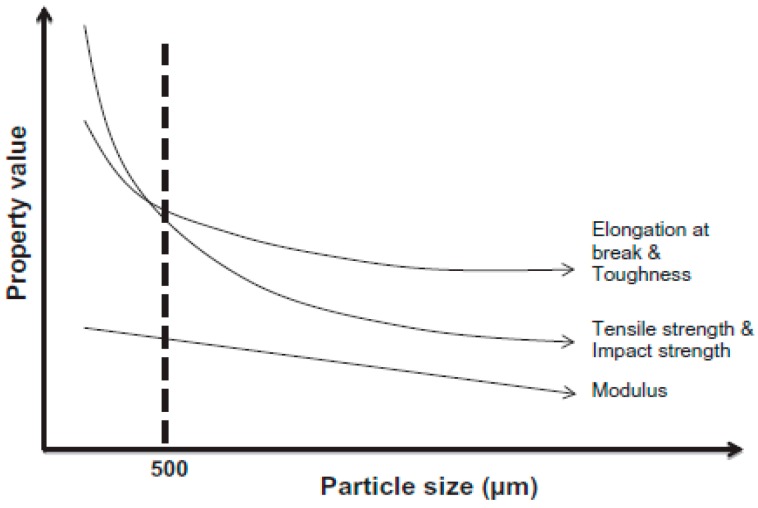
Effect of ground tire rubber (GTR) particle size on the mechanical properties of thermoplastic blends. Adapted with permission from [1]; copyright 2020 Elsevier Ltd.

**Figure 17 materials-13-00782-f017:**
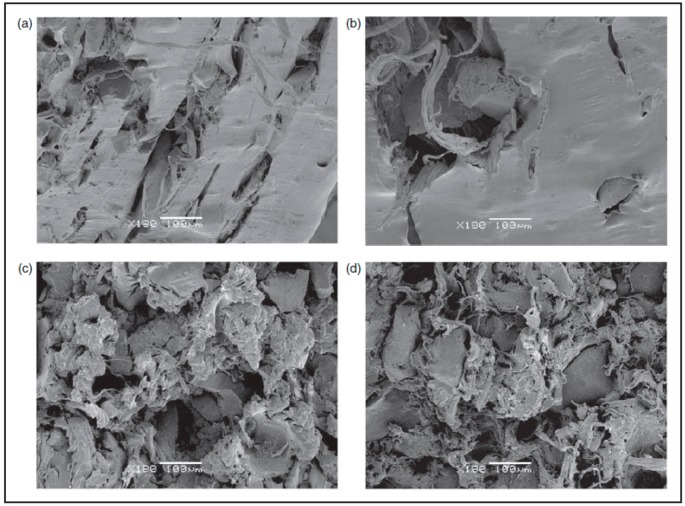
SEM micrographs of ethylene-vinyl acetate (EVA) blends with different ground tire rubber (GTR) contents: (**a**) 10 wt.%, (**b**) 20 wt.%, (**c**) 50 wt.%, and (**d**) 70 wt.%. Adapted with permission from [74]; copyright 2019 SAGE Publications Ltd.

**Figure 18 materials-13-00782-f018:**
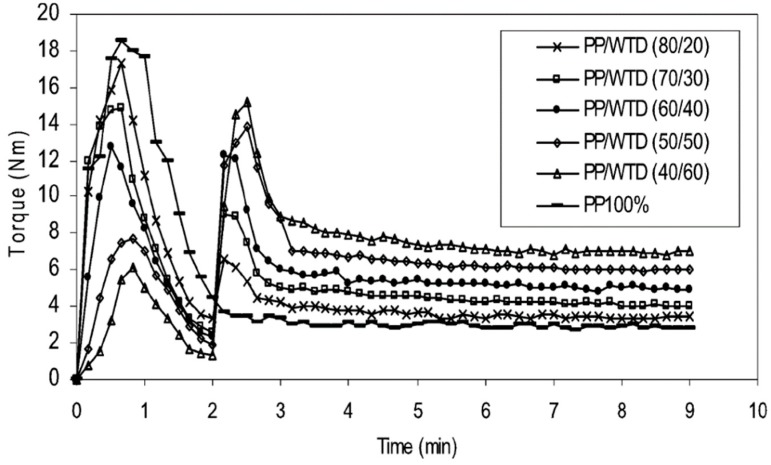
Torque evolution for polypropylene (PP)/waste tire dust (WTD) blends (250–500 μm). Adapted with permission from [73]; copyright 2019 Taylor & Francis Ltd.

**Figure 19 materials-13-00782-f019:**
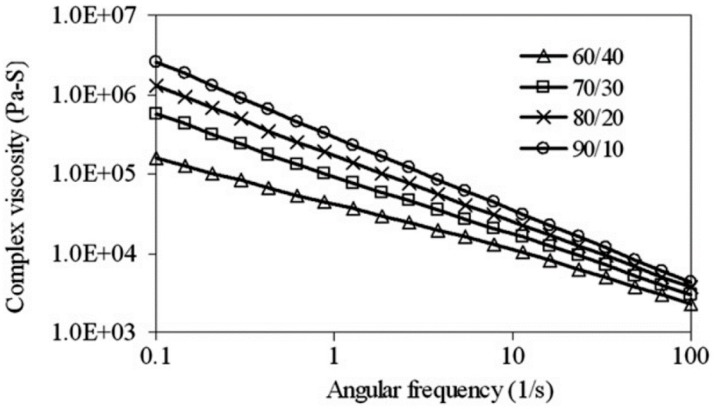
Complex viscosity as a function of angular frequency for thermoplastic natural rubber (TPNR) based on different natural rubber (NR)/high-density polyethylene (HDPE) ratios. Adapted with permission from [82]; copyright 2019 Elsevier Ltd.

**Figure 20 materials-13-00782-f020:**
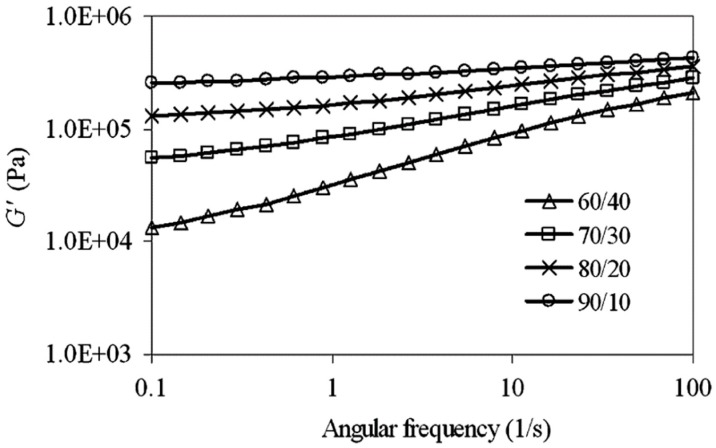
Storage modulus (G′) as a function of angular frequency for thermoplastic natural rubber (TPNR) based on different natural rubber (NR)/high-density polyethylene (HDPE) ratios. Adapted with permission from [82]; copyright 2019 Elsevier Ltd.

**Figure 21 materials-13-00782-f021:**
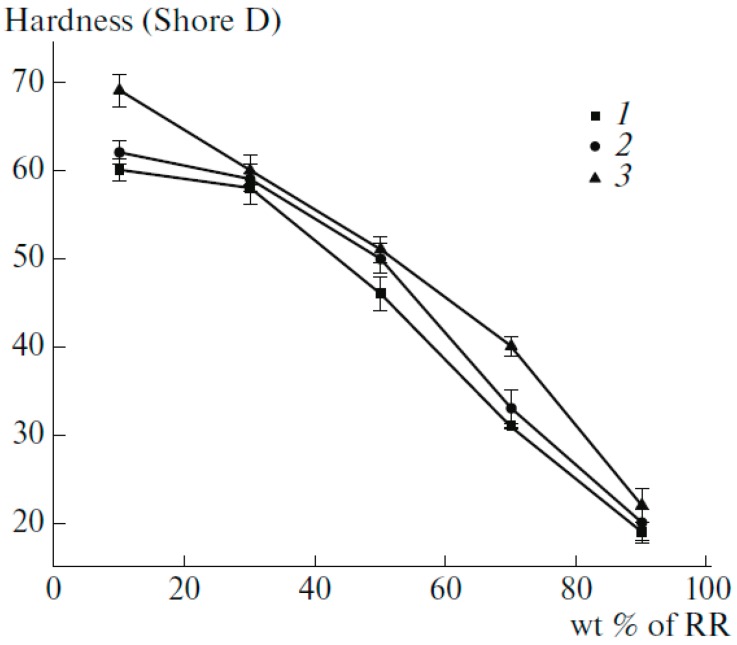
Hardness of high-density polyethylene (HDPE) as a function of reclaimed rubber (RR) content: (1) H-R10, (2) HR10-C, (3) H-R10-P. Adapted with permission from [77]; copyright 2019 Springer Nature Ltd.

**Figure 22 materials-13-00782-f022:**
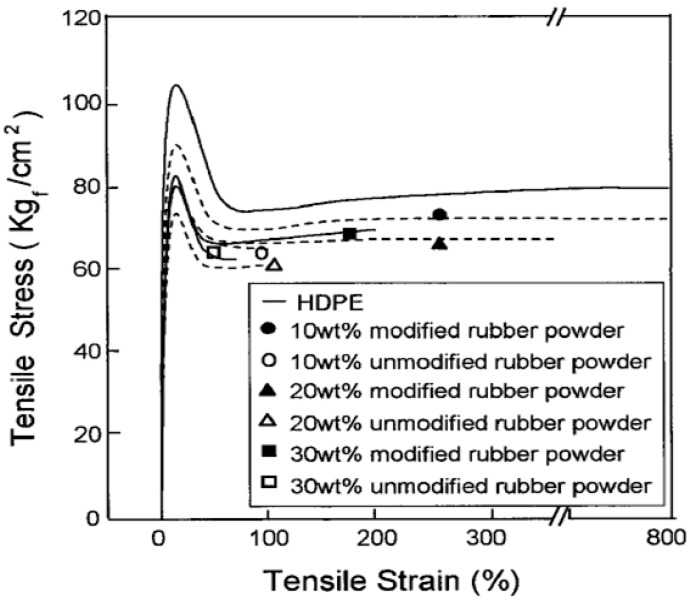
Tensile stress-strain curves of high-density polyethylene (HDPE) and HDPE/ground tire rubber (GTR) compounds. Adapted with permission from [50]; copyright 2019 John Wiley and Sons Ltd.

**Figure 23 materials-13-00782-f023:**
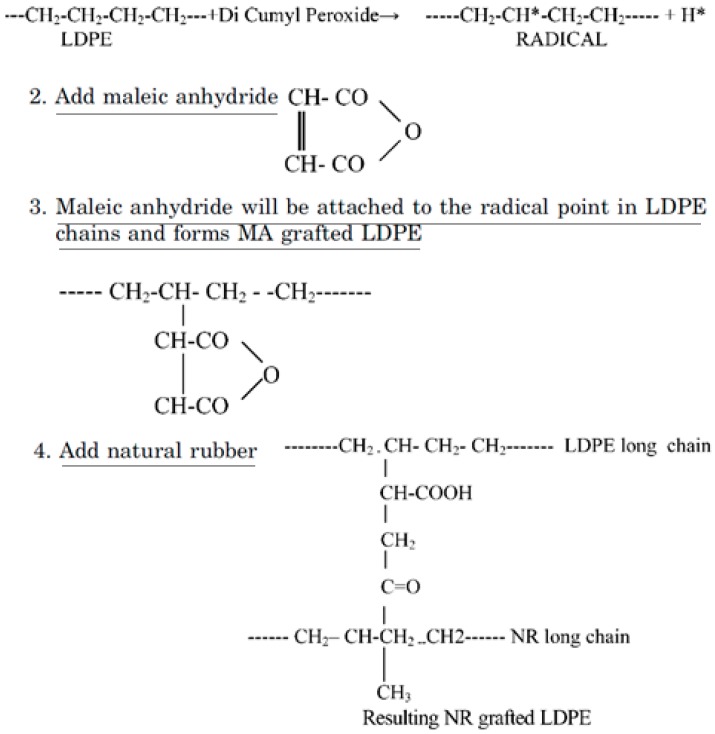
Reaction mechanism for low-density polyethylene (LDPE)/natural rubber (NR) modified with maleic anhydride (MA). Adapted with permission from [85]; copyright 2019 Taylor & Francis Ltd.

**Figure 24 materials-13-00782-f024:**
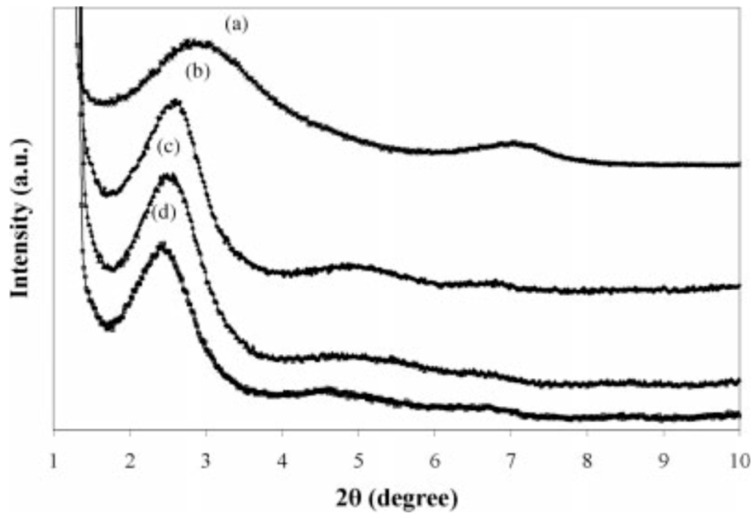
X-ray diffraction patterns of: (**a**) Cloisite 15A and TPE nanocomposites based on polypropylene (PP) with: (**b**) 60%, (**c**) 40%, and (**d**) 20% ethylene-propylene-diene monomer (EPDM). Adapted with permission from [87]; copyright 2019 John Wiley and Sons Ltd.

**Figure 25 materials-13-00782-f025:**
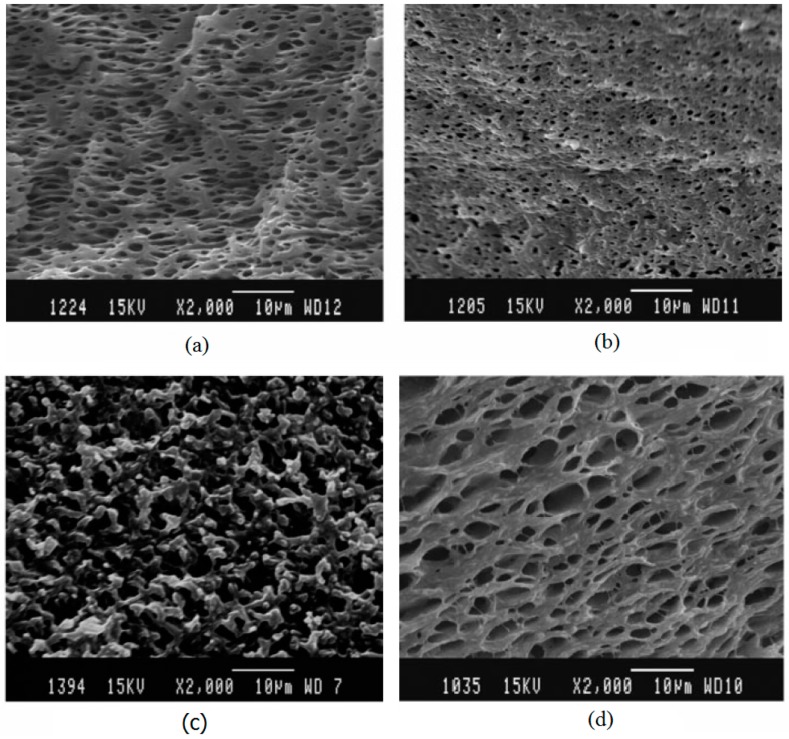
SEM micrographs of TPE based on: (**a**) unfilled polypropylene (PP)/ethylene-propylene-diene monomer (EPDM) (60/40), (**b**) nanoclay-filled PP/EPDM (60/40), (**c**) unfilled PP/EPDM (40/60), and (**d**) nanoclay-filled PP/EPDM (40/60) blends. Adapted with permission from [87]; copyright 2019 John Wiley and Sons Ltd.

**Figure 26 materials-13-00782-f026:**
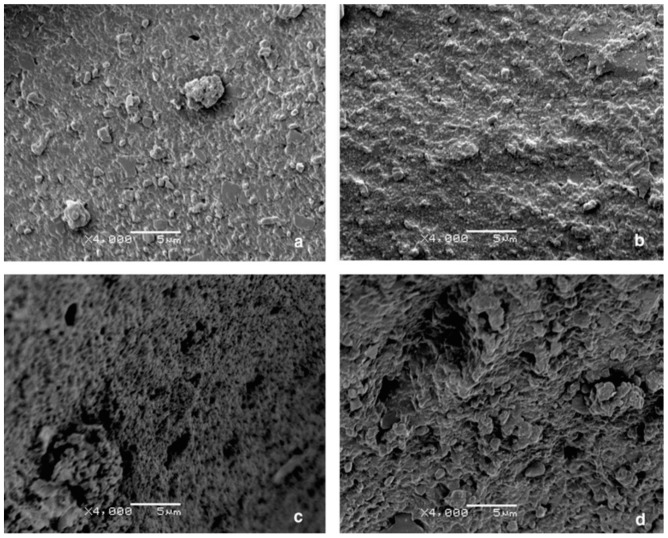
SEM of the ground tire rubber (GTR) particles surface: (**a**) untreated and treated with: (**b**) perchloric acid (HClO_4_), (**c**) nitric acid (HNO_3_), and (**d**) sulphuric acid (H_2_SO_4_). Adapted with permission from [89]; copyright 2019 Elsevier Ltd.

**Figure 27 materials-13-00782-f027:**
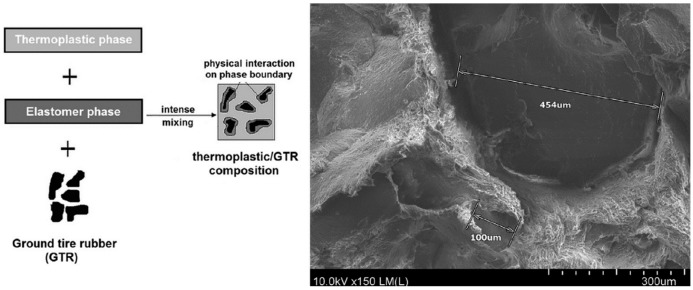
Compatibilisation mechanism of thermoplastic/ground tire rubber (GTR) blends using an elastomer as a modifier. Adapted with permission from [80]; copyright 2019 Elsevier Ltd.

**Figure 28 materials-13-00782-f028:**
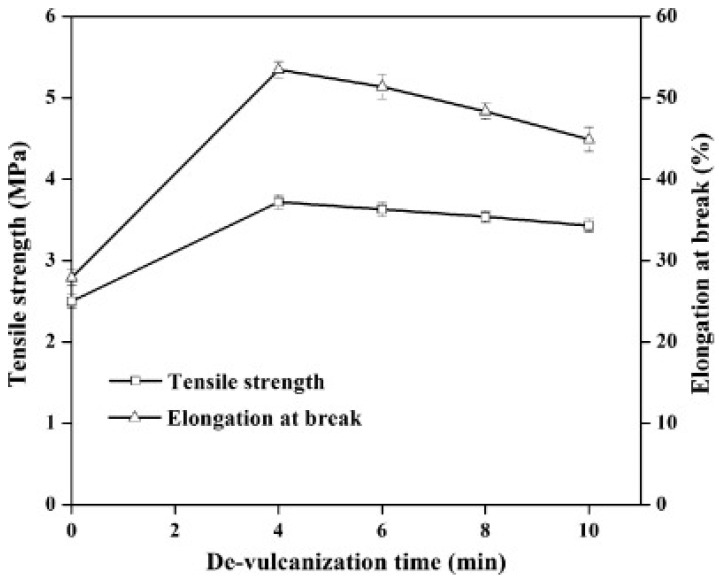
Tensile strength and elongation at break of dynamically cured devulcanized rubber (DR)/copolyester (COPE) blends as a function of the devulcanisation time at 180 °C. Adapted with permission from [91]; copyright 2019 Elsevier Ltd.

**Figure 29 materials-13-00782-f029:**
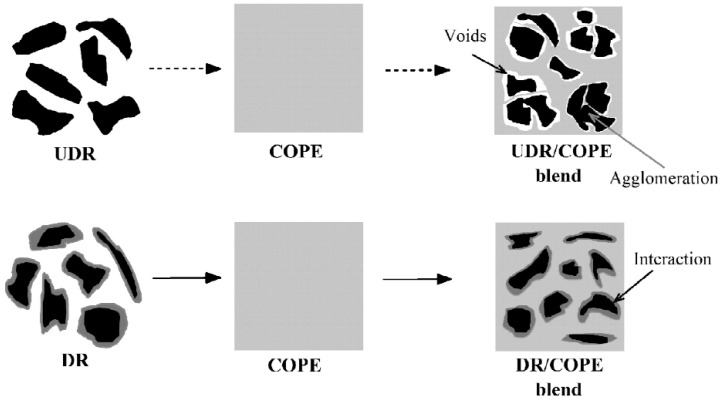
Schematic representation of the microstructure differences between the thermoplastic vulcanizates (TPV) based on devulcanized rubber (DR)/copolyester (COPE) and undevulcanised rubber (DR)/copolyester (COPE) blends. Adapted with permission from [91]; copyright 2019 Elsevier Ltd.

**Table 1 materials-13-00782-t001:** Typical compositions of tires [1,2,21,22].

Material	Cars/Passenger (wt.%)	Trucks (wt.%)
Rubber	41–48	41–45
Carbon Black	22–28	20–28
Metal	13–16	20–27
Textile	4–6	0–10
Additives	10–12	7–10

**Table 2 materials-13-00782-t002:** General methods of waste tire downsizing [1,2,22].

Methods	Description	Advantages	Disadvantages
Ambient (0.3 mm rough, irregular)	Repeated grinding following shredder, mills, knife, granulators, and rolling mills	High surface area and volume ratio	Temperature could rise up to 130 °C
			Oxidation on the surface of granulates
			Cooling needed to prevent combustion
Wet ambient (100 μm rough, irregular)	Grinding suspension of shredded rubber using grindstone	Lower level of degradation on granulates	Requires drying step and shredding of tires before grinding
	Water cools granulates and grindstone	High surface area and volume	
Water jet (rough, irregular)	Used for large sized tires (trucks and tractors)	Environmentally safe, energy saving, low level of noise, and no pollutants	Requires high pressure and trained personnel
	Water jet of >2000 bar pressure and high velocity used to strip rubber		
Berstoff’s method (rough, irregular)	Combines a rolling mill with a specially designed twin screw extruder in a line.	Small grain size, large specific area, and low humidity	Not disclosed
Cryogenic (75 μm sharp edge flat/smooth)	Rubber cooled in liquid nitrogen and shattered using impact type mill	No surface oxidation of granulates and cleaner granulates	High cost of liquid nitrogen
			High humidity of granulates

**Table 3 materials-13-00782-t003:** Energy required for cleaving typical bonds in vulcanized rubbers [71].

Type of Bond	Energy Required for Cleavage (kJ/mol)
C–C	348
C–S–C	285
C–S–S–C	268
C–S_x_–C	251

**Table 4 materials-13-00782-t004:** Tensile properties of high-density polyethylene (HDPE)/reclaimed rubber (RR) blends. Adapted with permission from [77]; copyright 2019 Springer Nature Ltd.

Sample Code	Tensile Strength (MPa)	Tensile Modulus (MPa)	Elongation at Break (%)
H-R30	11.0 ± 0.1	166.7 ± 7.3	31.5 ± 2.7
H-R50	6.0 ± 0.6	101.4 ± 2.8	61.3 ± 5.5
H-R70	2.3 ± 0.1	26.3 ± 5.8	125 ± 6.2
H-R90	0.6 ± 0.2	1.6 ± 0.7	149 ± 4.3
H-R30-C	12.2 ± 4.1	218.4 ± 6.1	45.1 ± 5.7
H-R50-C	7.3 ± 3.2	122.4 ± 3.8	78.9 ± 7.1
H-R70-C	3.0 ± 0.7	29.8 ± 1.8	138.6 ± 2.4
H-R90-C	0.9 ± 0.2	2.1 ± 0.9	183 ± 4.9
H-R30-P	13.5 ± 5.1	346.6 ± 7.4	58 ± 8.2
H-R50-P	9.4 ± 2.6	184.9 ± 5.8	94.1 ± 3.4
H-R70-P	6.0 ± 1.7	36.5 ± 2.7	152.2 ± 1.5
H-R90-P	2.7 ± 0.8	2.8 ± 1.5	213.5 ± 6.1

H: HDPE, R: reclaimed rubber, C: compatibiliser (PE-g-MA), P: peroxide (liquid peroxide with trade name DHBP (2,5-dimethyl-2,5-di-(tert-butylperoxy)-hexane)).

**Table 5 materials-13-00782-t005:** Hardness (Shore A) of low-density polyethylene (LDPE)/natural rubber (NR) and LDPE/nitrile-butadiene rubber (NBR) with and without particle modification. Adapted with permission from [85]; copyright 2019 Taylor & Francis Ltd.

LDPE/NBR	LDPE/NR	Ungrafted	Acrylic Acid (AA) Grafted	Maleic Anhydride (MA) Grafted
-	80/20	90	97	95
-	60/40	82	90	85
-	40/60	65	70	69
-	20/80	55	57	58
80/20	-	95	97	98
60/40	-	85	86	87
40/60	-	65	70	72
20/80	-	55	58	56

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
