# Peer review of "Waste Rubber Recycling: A Review on the Evolution and Properties of Thermoplastic Elastomers"

_materials, 2020, doi:10.3390/ma13030782_

Round 1
Reviewer 1 Report
The manuscript reports an overview of the most relevant aspects involving the rubber recycling subject. In general, the text is easy to read, well written with a good level of English, and the number of references is sufficient for such work.
Prior to publication I suggest:
Correct the number of the affiliation in the Authors name, line 3- Increase the quality of the figures, Table 3: reduce the size of the text under the table. - Correct the references issue " (Error! Reference source not found.") at lines 570, 576, 603, 614, 634-637, 670, 690, 695.Author Response
Thank you for the comments.
Typos were corrected.
Reviewer 2 Report
This manuscript provides a good (and very readable) guide to rubbery polymers. The manuscript is dominated by long sections of introductory text relating to various rubbers, but the 'recycling' aspect to the manuscript is rather limited. For example, there is a short section (section 2) at the end of page five and extends for about a single page. The manuscript then continues to provide background to other rubbers. Recycling is not mentioned again until page 12 (section 5.1), but again this is limited to a paragraph. The discussion of recycling seems to focus on the use of reclaimed rubber in other polymeric systems, but the emphasis does not seem to be on 'recent' as indicated by the title. A superficial glance at the reference list would perhaps support this as a significant number of references could not be classed as 'recent'. In this sense, this manuscript has the feel of a book chapter rather than a review paper.
In essence, the use of the word 'recent' needs to be justified through the critical analysis of more recent literature in the area and the introductory (textbook) content scaled back. Also, there are referencing issues - see the Error statements in the body of the text.
Author Response
This is the same as reviewer #1 !!!
The corrections have been made.
Reviewer 3 Report
Recent Advances…
Rubber recycling is an important research field in order to found a method to reuse polymer waste materials.
The authors, in the case of rubber recycling, consider the most convenient way that can go through size reduction by shredding /grinding; with this method it can be produced powders with different size and shape. It is opinion of the reviewer that the authors could indicate the most convenient size and shape for blending the recycled rubber with thermoplastic resins to produce thermoplastic elastomers (TPE) with technical characteristics similar to those of the market. In particular the work is focused on the recycle rubber that was grafted in order to produce the not identified TPE. In the introduction the authors examine the rubber types: in this contest the reviewer suggests to the authors, in the case of the vulcanization of EPM/EPDM to add a reference to the following up to date papers:
Milani/Milani – J. Math. Chem. – 49, 1357 – 2011 Milani/Milani – J. Appl. Poly. Sci – 124, 311 – 2012 Van Duin et al. –Rub. Chem. Tech. – 2017In relation to the table 2 where it is shown the energy required for cleavage in function of the bond type, the reviewer asks to the authors Why the energy of C – Sx – C is higher than the cleavage energy of C – S – S – C ? The other question is: How is it possible to obtain a commercial TPV starting from EPM/EPDM vulcanized and by grinding and PP? In the figure 7 it is shown the microscope morphology of a block copolymer thermoplastic elastomer but for the reviewer this figure is not clear. Can the authors change?
The TPO is different by TPV: What is the strategy in order to obtain these different products starting from rubbers vulcanised?
In Section 5 or rubber modification, the authors write that, in order to adapt any rubber to blend with a lot of plastic materials, it is needed to modify the rubber by grafting; what is the best process?
Author Response
Thank you for the comments
In relation to the table 2 where it is shown the energy required for cleavage in function of the bond type, the reviewer asks to the authors Why the energy of C – Sx – C is higher than the cleavage energy of C – S – S – C ?
The bonds energy presented in the manuscript (table 2) were taken from a review paper [1]. After double checking the original reference reporting on the bonds energy values, it was found that these values were wrong (thank you for pointing this out). The table was modified to show the corrected energy values required for cleaving typical bonds in vulcanized rubbers. In fact, devulcanization requires high energy to break the –C–S-C– (285 kJ/mol), –C–S–S–C– (268 kJ/mol) or –C–Sx–C– (251 kJ/mol) bonds [2]. It is seen that Sx bond energy is the lowest bond energy in the vulcanized rubber. The difference between the C–C main chain bond energy and the bond energy of the weakest bonds in the crosslinks (Sx) is equal to 97 kJ/mol. The difference between bond energies of the main chain C–C bonds and bond energies for the strongest crosslink is 63 kJ/mol. Specific level of energy is essential to break the main chain bonds and intermolecular bonds while the main chain bonds breaking requires higher energy levels than those required to breakup crosslinks [3].
How is it possible to obtain a commercial TPV starting from EPM/EPDM vulcanized and by grinding and PP?
Nowadays, PP/EPDM TPVs have gained significant attention for automotive applications. The most widely used compositions are based on dynamically vulcanized EPDM blended with a polyolefin (mainly PP). The preparation of these TPVs takes place at 180-210oC at 90-120 rpm. Twin-screw extruders are more efficient due to their better mixing. Dynamic vulcanization can be achieved by the incorporation of around 8 parts of sulfur per 100 parts of rubber (phr) at 140oC for about 5 h. However, the addition of zinc oxide decreases the vulcanization time to less than 3 h. The use of accelerator as low as 0.5 phr can significantly reduce this time, so dynamic vulcanization by sulfur needs an accelerator to have commercial benefits. Accelerated sulfur vulcanization of PP/EPDM is performed in the presence of activators (ZnO and steric acid) and accelerators such as tetramethylthiuram disulfide (TMTD) and dibenzothiazole disulfide (MBTS). The dynamic vulcanizate based on the blend of EPDM and PP displays a disperse morphology. This morphology is known to be independent of the elastomer-thermoplastic ratio or the molecular weights of the polymers. As the average particle diameter of vulcanized EPDM particles decreases, the tensile strength and elongation at break increase [4].
In the figure 7 it is shown the microscope morphology of a block copolymer thermoplastic elastomer but for the reviewer this figure is not clear. Can the authors change?
The figure was changed.
The TPO is different by TPV: What is the strategy in order to obtain these different products starting from rubbers vulcanised?
In TPV blends, the crosslinked elastomer particles are dispersed in the thermoplastic matrix. The polyolefin phase in a TPV is continuous enclosing the crosslinked elastomeric phase. The vulcanized rubber particles need to be partially devulcanized to destroy the crosslinked structure and have sufficient interaction (entanglements) with the thermoplastic chains. In the vulcanized rubber, free chains on the surface of devulcanized rubber promote co-crosslinking between the devulcanized rubber and the polymer matrix through dynamic vulcanization. In fact, the breakdown of the rubber crosslinked structure decreases the crosslink density giving better ability to form networks by dynamic vulcanization. In general, devulcanized rubber particles show higher chain mobility. On the other hand, in TPO blends the elastomeric phase does not need to be crosslinked. The rubber phase may be either continuous or discrete, depending on the amount of rubber relative to the amount of polyolefin, the type of rubber, and the mixing procedure. The simple TPO blends can be prepared by mixing the hard polymer and the soft elastomer together in high shear compounding equipment, such as batch mixers (Banbury) or continuous mixers (single- or twin-screw extruders). The crosslinked gel content of the ground tire rubber (GTR) particles with an inability to be dispersed in continuous matrix act as stress concentrating point limiting molecular orientation and mobility. Therefore, increasing the GTR content in TPE blends increases the gel content resulting in lower tensile strength and elongation at break. GTR devulcanization, breaking the crosslinked structure by the rupture of sulfur bonds (S-S or C-S), leads to produce a soluble fraction which is responsible for the interaction with the thermoplastic resin improving the TPE mechanical properties. However, nonselective breakup of the crosslinked structure during devulcanization might result in the scission of C-C bonds in the main backbone chains leading to a drop of molecular weight and TPE mechanical properties [5].
In Section 5 or rubber modification, the authors write that, in order to adapt any rubber to blend with a lot of plastic materials, it is needed to modify the rubber by grafting; what is the best process?
Grafting monomers such as styrene, allylamine, acrylamide and methacrylate onto rubber particles through free-radical initiation or photo-initiation prevents particles agglomeration and leads to achieve smaller particle size and more homogeneous distribution in the continuous polymer matrix to achieve better blend properties. Also, unsaturated monomers grafted on the rubber surface can participate in the co-crosslinking reaction or entangle with the matrix macromolecules, as well as improving the compatibility of the polar grafted monomers with the polymer matrix by incorporating reactive compatibilizer such as maleic anhydride grafted polymer. Grafting monomers onto rubber particles in the presence of solvent containing monomers can be carried out through free-radical initiation or photo-initiation. Benzoyl and dicumyl peroxide are the most common free radical initiators and high energy electron, like X-rays, UV or visible light are used for photo-grafting initiation. The modification approach can decrease the agglomeration of GTR particles while the compatibility of the polar monomer grafted GTR with the polymer matrix can be further improved by incorporating reactive compatibilizers such as maleic anhydride grafted polymer [1].
References
Ramarad, S.; Khalid, M.; Ratnam, C.; Chuah, A.L.; Rashmi, W. Waste tire rubber in polymer blends: A review on the evolution, properties and future. Progress in Materials Science 2015, 72, 100-140. Diao, B.; Isayev, A.; Levin, V.Y. Basic study of continuous ultrasonic devulcanization of unfilled silicone rubber. Rubber chemistry and technology 1999, 72, 152-164. Rooj, S.; Basak, G.C.; Maji, P.K.; Bhowmick, A.K. New route for devulcanization of natural rubber and the properties of devulcanized rubber. Journal of Polymers and the Environment 2011, 19, 382-390. Kajzar, F.; Pearce, E.M.; Turovskij, N.A.; Mukbaniani, O.V. Key Engineering Materials: Interdisciplinary Concepts and Research; CRC Press: 2014; Vol. 2. Drobny, J.G. Handbook of thermoplastic elastomers; Elsevier: 2014.
Reviewer 4 Report
The paper needs a significant improvement:
English should be improved. Not only in terms of grammar but in style.
The structure of the article is not well organised. The choice of the main topics to be developed is somewhat confusing. Exemples:
the point recycling is followed by TPE. Titles as "compatibility" are not significatives of the concept that is developed later. Important concepts for recycling, as reclaiming are mixed with generalities A short point called recycling in a paper with this title is not appropriateIntroduction is too short and underdeveloped.
Conclusions as presented, could also be included in Introduction.
The paper includes very general concepts explained in many books and also very specific results. A good combination with these festures is very difficult to achieve. The reader of a journal as Materials expect some state of the art in the recycling of rubber more than general concepts, specially with a title as "Recent advances.... "
Author Response
Thank you !
The title was changed and the comment implemented in the revised version.
Round 2
Reviewer 2 Report
The manuscript seems to be improved in that there is greater emphasis on the tire aspect. The title change is acknowledged and now better represents the content. I feel that the manuscript represents a reasonable (and quite readable) review of the subject matter and can be accepted.
Author Response
Thank you !
Reviewer 3 Report
The authors implemented all the necessary corrections and the paper is now acceptable for publication.
Author Response
Thank you !
Reviewer 4 Report
I think this paper has an structural issue. The distribution of the contents and the main structure of the text is confuse.
Authors mix very general concepts to very specific ones. After several generalistic points (with a division in sections that is very arguable) the point 6 start reviewing very specific papers. Usually in a review in a paper the reader expect the type of content included in point 6 but structured in a way that aspires to the organization of the knowledge adquired during the study. The main point of the review paper is to present in an ordered manner, the information of several authors working in fields that are related but that has been published dispersely. I think this is not achieved in the paper.
Author Response
About the “structure problem” of the paper as mentioned by the reviewer, it should be mentioned that we tried to give informative and concise information on the different rubber types as well as their recycling difficulties in the 1st section of the article. In the 2nd section, waste tire recycling methods are presented and the melt blending of these waste materials with thermoplastics is reported as the most straightforward and environmentally friendly method. Actually, it was necessary to introduce different types of TPE compounds in section 3 and the production challenges (blend incompatibility) for these compounds in section 4. Next, several papers related to the improvement of TPE blend compatibility are presented in section 5 and 6 to present the different approaches to improve the TPE blends properties.
To improve on this, a part was added at the beginning of the manuscript and more sub-section titles were are added to better define the structure.